# COVID-19 and common mental health symptoms in the early phase of the pandemic: An umbrella review of the evidence

Anke B. Witteveen[1]*, Susanne Y. Young[1,2], Pim Cuijpers[1], José Luis Ayuso-Mateos[3,4], Corrado Barbui[5], Federico Bertolini[5], Maria Cabello[3,4], Camilla Cadorin[5], Naomi Downes[6], Daniele Franzoi[1], Michael Gasior[1], Brandon Gray[7], Maria Melchior[6], Mark van Ommeren[7], Christina Palantza[1], Marianna Purgato[5], Judith van der Waerden[6], Siyuan Wang[1], Marit Sijbrandij[1]

1 Department of Clinical, Neuro- and Developmental Psychology, Amsterdam Public Health Institute and World Health Organization Collaborating Center for Research and Dissemination of Psychological Interventions, Vrije Universiteit, Amsterdam, the Netherlands, 2 South African PTSD Research Programme of Excellence, Department of Psychiatry, Faculty of Medicine and Health Sciences, Stellenbosch University, Stellenbosch, South Africa, 3 Department of Psychiatry, Universidad Autonoma de Madrid, WHO Collaborating Center for Research and Training in Mental Health Services at the Universidad Autónoma de Madrid, Madrid, Spain, 4 Centro de Investigación Biomédica en Red de Salud Mental, CIBERSAM, Instituto de Salud Carlos III, Madrid, Spain, 5 WHO Collaborating Centre for Research and Training in Mental Health and Service Evaluation, Department of Neuroscience, Biomedicine and Movement Sciences, University of Verona, Verona, Italy, 6 Sorbonne Université, INSERM, Institut Pierre Louis d'Epidémiologie et de Santé Publique, Equipe de Recherche en Epidémiologie Sociale, Paris, France, 7 World Health Organization, Department of Mental Health and Substance Use, Geneva, Switzerland

* a.b.witteveen@vu.nl

**Data Availability Statement:** All relevant data are within the manuscript and its Supporting Information files.

## Abstract

### Background

There remains uncertainty about the impact of the Coronavirus Disease 2019 (COVID-19) pandemic on mental health. This umbrella review provides a comprehensive overview of the association between the pandemic and common mental disorders. We qualitatively summarized evidence from reviews with meta-analyses of individual study-data in the general population, healthcare workers, and specific at-risk populations.

### Methods and findings

A systematic search was carried out in 5 databases for peer-reviewed systematic reviews with meta-analyses of prevalence of depression, anxiety, and post-traumatic stress disorder (PTSD) symptoms during the pandemic published between December 31, 2019 until August 12, 2022. We identified 123 reviews of which 7 provided standardized mean differences (SMDs) either from longitudinal pre- to during pandemic study-data or from cross-sectional study-data compared to matched pre-pandemic data. Methodological quality rated with the Assessment of Multiple Systematic Reviews checklist scores (AMSTAR 2) instrument was generally low to moderate. Small but significant increases of depression, anxiety, and/or general mental health symptoms were reported in the general population, in people with pre-existing physical health conditions, and in children (3 reviews; SMDs ranged from 0.11 to

**Funding:** This work is supported by the World Health Organization (WHO, https://www.who.int/) [grant to MS] and the European Union's Horizon 2020 - Framework Programme for Research and Innovation Societal changes (2014–2020) (https://www.eeas.europa.eu/eeas/horizon-2020_en) [grant agreement no. 101016127 to MS]. The funders had no role in study design, data collection and analysis, decision to publish, or preparation of the manuscript.

**Competing interests:** The authors have declared that no competing interests exist.

**Abbreviations:** COVID-19, Coronavirus Disease 2019; LMIC, low- and middle-income country; PTSD, post-traumatic stress disorder; SMC, standardized mean change; SMD, standardized mean difference; WHO, World Health Organization.

0.28). Mental health and depression symptoms significantly increased during periods of social restrictions (1 review; SMDs of 0.41 and 0.83, respectively) but anxiety symptoms did not (SMD: 0.26). Increases of depression symptoms were generally larger and longer-lasting during the pandemic (3 reviews; SMDs depression ranged from 0.16 to 0.23) than those of anxiety (2 reviews: SMDs 0.12 and 0.18). Females showed a significantly larger increase in anxiety symptoms than males (1 review: SMD 0.15). In healthcare workers, people with preexisting mental disorders, any patient group, children and adolescents, and in students, no significant differences from pre- to during pandemic were found (2 reviews; SMD's ranging from −0.16 to 0.48). In 116 reviews pooled cross-sectional prevalence rates of depression, anxiety, and PTSD symptoms ranged from 9% to 48% across populations. Although heterogeneity between studies was high and largely unexplained, assessment tools and cut-offs used, age, sex or gender, and COVID-19 exposure factors were found to be moderators in some reviews. The major limitations are the inability to quantify and explain the high heterogeneity across reviews included and the shortage of within-person data from multiple longitudinal studies.

## Conclusions

A small but consistent deterioration of mental health and particularly depression during early pandemic and during social restrictions has been found in the general population and in people with chronic somatic disorders. Also, associations between mental health and the pandemic were stronger in females and younger age groups than in others. Explanatory individual-level, COVID-19 exposure, and time-course factors were scarce and showed inconsistencies across reviews. For policy and research, repeated assessments of mental health in population panels including vulnerable individuals are recommended to respond to current and future health crises.

## Author summary

### Why was this study done?

The Coronavirus Disease 2019 (COVID-19) pandemic has been one of the greatest global public health challenges of the last century and has impacted multiple aspects of health and public life.

An adverse association between the pandemic and global mental health was expected and many research projects have been rapidly developed to assess this.

There is uncertainty about the degree and extent of the associations between the pandemic and its associated measures and mental health.

### What do these findings mean?

This umbrella review could help clinicians, researchers, and policy makers to better understand the current evidence on the association between the COVID-19 pandemic and mental health, particularly in specific vulnerable subpopulations.

The interpretability of the included systematic reviews was limited by the great variation in prevalence rates and associations between studies and because of the scarcity of longitudinal data.

Policy makers and researchers should address common pitfalls of research designs prior to implementation of systematic mental health assessments in future population panels.

## What did the researchers do and find?

We synthesized evidence from 123 systematic reviews of individual studies on symptoms of common mental disorders, including depression, anxiety, and PTSD, in general and specific populations and in healthcare workers. Seven reviews compared differences in mental health outcomes during the COVID-19 pandemic or during implementation of public health and social measures to pre-pandemic periods or periods with minimal restrictions. Another 116 reviews provided combined data on during pandemic prevalence rates of mental health outcomes.

Mental health and particularly mood of the general population slightly deteriorated in the first half year of the pandemic, and symptom increases were associated with periods of public health measures and social restrictions. Also, people with preexisting physical health conditions, females, and young people showed pandemic-associated increases in symptoms.

Variation in pandemic-associated mental health prevalence rates between individual studies was large and often unexplained. In several reviews, methodological, individual-level, and COVID-19 exposure factors did explain some of the variation but in others this was not the case. Quality of the systematic reviews was poor to moderate.

## Introduction

The Coronavirus Disease 2019 (COVID-19) pandemic has led to world-wide human suffering. Besides the physical impact, COVID-19 disease may have a direct mental health impact [1,2] as well as an indirect psychological impact through implementation of public health measures and social restrictions and its longer-term socioeconomic consequences [3]. Although findings from population-based studies in the initial stages of the pandemic indicate that most people were resilient and did not experience increases in distress [4], findings also suggest an increase of common mental health symptoms such as depression and anxiety symptoms [5]. Specifically vulnerable populations such as people dealing with financial problems, suffering from poverty, being from ethnic or racial minorities, or having preexisting health conditions have been challenged more than others, both in terms of infection and death rate from COVID-19 disease and in terms of mental health impact [4,6,7].

Since the start of the pandemic, the evidence base on the association between COVID-19 and mental health has evolved rapidly. A large number of cross-sectional and longitudinal studies have been published assessing associations between mental health and COVID-19 across the general population and vulnerable groups. These studies have been integrated in numerous systematic reviews and meta-analyses and suggest a significant adverse association between the pandemic and mental health mainly by presenting pooled prevalence rates from cross-sectional studies. However, the few reviews of longitudinal pooled data show less pronounced increases or are contradictory in terms of association between mental health symptoms and the pandemic, for example, in subgroups [8–10]. The main difficulty in getting a more accurate picture of the association between COVID-19 and mental health, is that many studies included in these reviews have methodological weaknesses. Importantly, most individual studies have cross-sectional designs and lack pre- to during pandemic longitudinal data,

which makes causal inferences to the pandemic difficult. Furthermore, even when multiple during- and pre-pandemic assessments have been performed in surveys, respondents often come from nonrepresentative convenience samples while a probability sampling approach is lacking [11]. Assessments of mental health outcomes are often not based on structured clinical interviews or validated questionnaires with established cut-offs and methodological quality or risk of bias assessment of primary studies is often lacking. These shortcomings lead to conflicting conclusions, and therefore to confusion among policy makers and clinicians [12].

A more comprehensive overview of the large amount of meta-analyses of pooled estimates of mental health problems during the COVID-19 pandemic may increase further understanding of the relation between COVID-19 and mental health. A critical evaluation of the current research evidence is needed to correctly inform the global mental health response to mitigate (future) disruptions and to adapt research strategies and implementation of interventions to address COVID-19–related mental health problems in populations where needed most, such as in young people [13,14]. We aimed to provide an overview of the evidence base on common mental health disorder symptoms during the pandemic, ideally compared to pre-pandemic periods, using an umbrella review approach. With this qualitative approach, inconsistencies and gaps of knowledge in the evidence may be recognized [15–18].

The aim of this umbrella review was to integrate the findings of separate reviews with meta-analyses on the prevalence of mental health affected by the pandemic in the general population and in populations at risk for increased psychological distress related to the pandemic, such as healthcare workers, people with preexisting physical or mental conditions, patients with COVID-19 infections, and young people. In addition to providing a more accurate and complete picture of the association between the COVID-19 pandemic and mental health, we aimed to identify gaps in knowledge for further scientific research and to identify targets for clinical and policy interventions.

## Methods

### Umbrella review design

We followed guidelines for umbrella reviews [12,17,19]. This umbrella review was part of a broader umbrella review registered with a protocol in the Open Science Framework platform (https://doi.org/10.17605/OSF.IO/JF4Z2) developed to collate evidence on mental health impact of COVID-19 in a scientific brief of the World Health Organization (WHO) [20]. This study was reported as per the Preferred Reporting Items for Systematic Reviews and Meta-Analyses (PRISMA) guideline [21] (**S1 PRISMA** **Checklist**).

### Literature search strategy and eligibility criteria

A systematic search was carried out in Ovid MEDLINE All, Embase (Ovid), PsycINFO (Ovid), CINAHL, and Web of Science published between December 31, 2019 until October 6, 2021, using a general search string for mental health and COVID-19 by combining a broad range of text and keywords for COVID-19 pandemic and mental health and mental disorders (see **S1 Text** for the search strings). An update of the search was performed between October 7, 2021 and August 12, 2022. Couples of independent researchers (SY and FB, SW and MC, CP and CC, DF and JW, ND and MG) screened titles and abstracts independently with use of software tool Rayyan and Endnote for deduplication. Full texts of eligible records were screened by 2 independent researchers. Disagreements were resolved via discussion and consensus, involving a third, senior team member. Papers were included based on the following eligibility criteria: (1) published in a peer-reviewed international journal; (2) included study selection criteria; (3) systematically searched at least 1 bibliographic database; (4) included a list and

synthesis of included studies; (5) included primary studies with longitudinal cohort- or cross-sectional data or data from time-series designs (studies including other designs as well were only eligible when results were synthesized separately); (6) included primary studies with data collected after December 31, 2019 (first WHO report of Chinese outbreak [22]) in the general population, healthcare workers, or vulnerable groups such as people who have experienced "severe" Severe Acute Respiratory Syndrome Coronavirus 2 (SARS-CoV-2) infections, people with post-COVID condition, specific mental disorders or living in psychiatric institutions, children, adolescents and young adults (e.g. students) or people at risk due to being marginalized (e.g., on race/ethnicity, sex, or gender) or due to chronic medical conditions; and (7) reported the following outcomes (a) from longitudinal cohort or time series studies, i.e., a standardized mean change (SMC) or difference in prevalence of any mental disorder (excluding substance related and addictive disorders, degenerative neurological disorders, and sleep disorders) based on a validated diagnostic interview, change in proportion of participants above a cut-off on a validated mental health symptom questionnaire, or change in scores on a validated mental health questionnaire at multiple time points during COVID-19 or compared to pre-COVID-19 outcomes; or (b) from cross-sectional studies, i.e., prevalence of any mental disorder based on validated diagnostic interview or proportion of participants above a cut-off on a validated mental health symptom questionnaire on a single time point. There were no language restrictions. Reviews from other infectious disease epidemics were only eligible if they also included separate data from COVID-19 studies. During the course of the development of this umbrella review, we deviated from the a priori protocol in a few instances. First, we decided not to include reviews with meta-analyses published online in 2020 because of the multitude of reviews available and because most primary studies from 2020 will have been included in reviews from 2021 and 2022. Second, instead of critically assessing systematic reviews from non-peer–reviewed data repositories before inclusion, we excluded non-peer–reviewed systematic reviews because of potential bias due to quality issues or invalid results. Third, although our protocol mentions Chinese search terms, these have been omitted since we did not search Chinese databases (e.g., Wanfang). Fourth, the AMSTAR 2 classification of ratings based on critical and non-critical criteria mentioned in the protocol was adapted in the process of quality assessment for reasons explained below and in **S1 Text** (see note Table C in S1 Text).

## Quality assessment

Included reviews were rated by 2 independent assessors for their quality using the Assessment of Multiple Systematic Reviews checklist (AMSTAR 2) [23]. The 16-item AMSTAR 2 considers the quality of the search, description of individual studies, assessment of publication bias, use of appropriate statistical methods, assessment of risk of bias in individual studies, and reporting of sources of funding and conflicts of interest. The items were scored as No (0 points), Partial yes (0.5 points), or Yes (1 point). Discrepancies were resolved by consensus and after discussion with another reviewer in the team. Although the AMSTAR 2 authors put more emphasis on the critical item scores [23], this approach is debated and we therefore also calculated total scores for each individual systematic review included [24] (see Table C in **S1 Text**).

## Data extraction and synthesis

For included reviews, 2 researchers independently extracted name of the first author, publication year, number of primary studies included, sample size per pooled outcome, pooled prevalence of main outcomes, or statistics used in original paper with corresponding 95% CI (e.g., SMC or difference, Hedges' g, Cohens' d). Results of statistical tests for heterogeneity as well as narrative summaries of meta-regression results, subgroup or moderator analyses were

extracted as well (Tables A, B, and C in **S2 Text**). Individual study designs of meta-analyses (e.g., cross-sectional or longitudinal cohort, case-control) were also extracted, as well as countries or continents covered in each meta-analysis (**Table 1**). In case of disagreements between the 2, consensus was reached, including consultation of a third senior investigator as necessary. Data was extracted as reported in the reviews. The characteristics and major findings of the included reviews are presented using tables and figures.

## Results

### Characteristics of included studies

From the initial and updated searches, 77.758 records were retrieved (**Fig 1**). For the umbrella review, we identified 904 systematic reviews with or without meta-analyses. Of those, 781 reviews were excluded for several reasons such as not including a meta-analysis or wrong outcome (**Fig 1**), retaining 123 eligible reviews with meta-analyses of primary studies published in 2021 and 2022 (initial search [9,10,25–80]; updated search [5,81–144]).

Characteristics of the 123 included studies are provided in **Table 1**. Of the included studies, 44 performed meta-analyses assessing the association between COVID-19 and mental health symptoms in the general population, 103 in healthcare worker populations, and 68 in specific populations. The searches of the systematic reviews with meta-analyses covered the period up to and including the second or third quarters of 2020 (35 reviews), last quarter of 2020 (15 reviews), the first or second quarter of 2021 (51 reviews), the third and fourth quarter of 2021 (15 reviews), and 5 reviews searched for studies up to the first quarter of 2022 (2 not reported [39,125]). Of the 123 eligible review articles, the majority provided pooled prevalence rates for depression and anxiety (i.e., 108 and 101 reviews, respectively) and fewer for PTSD symptom levels (35 reviews). Meta-analyses included a mean of 43 primary studies and a median of 27 studies with a variety of designs. Only 7 systematic reviews either exclusively focused on providing pooled difference estimates based on longitudinal studies with during- and pre-pandemic assessments or also included cross-sectional studies with matched pre-pandemic or pre-implementation of public health and social measures prevalence data [5,10,43,59,89,98,113]. The majority of reviews reported pooled prevalence rates based on above cut-off values of validated measures from surveys or cohort studies with mainly cross-sectional designs (k = 116). Some of these reviews included longitudinal studies as well but without pooling the data. A range of countries and continents were covered by the meta-analyses although most studies were performed in China, United States of America, and Europe. Representation of individual studies from low- and middle-income countries (LMICs) (i.e., South America and Africa) was low (**Table 1**).

### Quality assessment of included studies

The AMSTAR 2 rated level of methodological quality assessment by outcome across all included systematic reviews and meta-analyses. **Fig 2** shows that, concerning the 7 critical domains of AMSTAR 2, an a priori protocol was established in 61% of systematic reviews with meta-analyses, 53% performed a comprehensive literature search, none of systematic reviews with meta-analyses provided a list of excluded studies with justification, 75% used satisfactory techniques for assessment of risk of bias in individual studies, 94% used appropriate methods for meta-analysis, 30% discussed risk of bias in interpretation of findings, and 71% investigated and discussed publication bias. Each AMSTAR 2 domain judgment for each included systematic review with meta-analyses is available in Table C in **S1 Text**. The total AMSTAR 2 score and sub-scores of critical items have been provided for each included review (**Table 1**). Total scores on AMSTAR 2 ranged from 2 to 13 with a mean total score of 8.5. Only the review of

**Table 1. Characteristics of included systematic reviews with meta-analyses.**

| Study | End date search | Studies | Sample size | Study designs | Assessment | Outcomes[1] | Study populations | Countries/continents/WHO regions | AMSTAR 2 scores | |
|---|---|---|---|---|---|---|---|---|---|---|
| | | | | | | | | | Critical | Total |
| Abdulla 2021 [81] | Feb 2021 | 23 | 8,855 | CS, OBS, L | Validated/unvalidated | Dep, Anx, | HCW | India | 3,5 | 9,5 |
| Adibi 2021 [25] | June 2020 | 19 | 21,866 | NR | Validated measures | Anx | HCW | China, USA, European countries, United Kingdom, South-Korea, Turkey, Brazil, India, Japan, Hong Kong, Singapore, Israel | 3 | 8 |
| Adrianto 2022 [82] | June 2021 | 54 | 95.326 | CS, CC, cohort, mixed | Validated | Dep | Pregnant, postpartum, perinatal | China, Hong Kong, Japan, Iran, Qatar, Israel, Egypt, Turkey, Italy, Switzerland, the Netherlands, Greece, Spain, UK, Ireland, Norway, Poland, USA, Canada, and Mexico | 2,5 | 9 |
| Afridi 2022 [83] | Jan 2022 | 10 | 12,507 | CS | Validated | Dep | HCW | Pakistan | 2,5 | 7,5 |
| Alzahrani 2022 [84] | August 2021 | 15 | 262,656 | CS | Validated/unvalidated | Dep, Anx | GP | Saudi | 2,5 | 6 |
| Arora 2022 [85] | April 2020 | 28 | 97,173 | CS, OBS | Validated/unvalidated | Dep, Anx, PTSD | HCW, GP, COVID Patients | China, Hong Kong, Italy, Iran, Vietnam, India, Singapore, USA, UK, | 1,5 | 7 |
| Aymerich 2022 [86] | March 2021 | 239 | 271,319 | CS | Validated | Dep, Anx, PTSD | HCW | Five continents: 150 (62.76%) from Asia, 55 (23.01%) from Europe, 20 (8.37%) from America, 11 (4.60%) from Africa, and 2 (0.84%) from Oceania; there was also 1 multicontinental study | 4,5 | 11 |
| Ayubi 2021 [26] | Jan 2021 | 21 | NR | CS, L, CC, OBS | Validated self-report | Dep, Anx | Patients (cancer) | South Korea, UK, China, USA, Europe, Turkey, Slovenia, Tunisia, Brazil, India, Japan, Germany, Hong Kong, Singapore, Italy, Israel, Poland, International, the Netherlands | 0,5 | 4,5 |
| Bello 2022 [87] | Sept 2021 | 78 | 62380 | NR | Validated/unvalidated | Dep, Anx | GP | Ethiopia, Nigeria, Egypt, Libya, South Africa, Ghana, Uganda, Morocco, Kenya, Tunisia, Libya, Cameroon, Zambia, Algeria, Togo, Sudan, Mali. Four studies covered more than 1 African country | 2,5 | 7,5 |
| Balakrishnan 2022 [88] | March 2021 | 82 | 201,953 | CS, L | Validated | Dep | HCW, students, GP | China, Japan, Hong Kong, South Korea, USA, Canada, Bangladesh, India, Nepal, Sri Lanka, Australia, Malaysia | 3,5 | 8 |
| Bareeqa 2021 [29] | Apr 2020 | 19 | 62,382 | CS | Validated self-report | Dep, Anx | GP, HCW | China | 3 | 8 |

*(Continued)*

**Table 1.** (Continued)

| Study | End date search | Studies | Sample size | Study designs | Assessment | Outcomes[1] | Study populations | Countries/continents/ WHO regions | AMSTAR 2 scores | |
|---|---|---|---|---|---|---|---|---|---|---|
| | | | | | | | | | Critical | Total |
| Batra 2021 [27] | July 2020 | 27 | 90,879 | OBS | Unspecified | Dep, Anx, PTSS | Students | China, Israel, Turkey, Jordan, USA, Italy, India, Albania, Brazil, Saudi Arabia, Greece, France, Russia, Belarus | 5 | 9 |
| Bussières 2021 [89] | June 2021 | 28 | 14209 | L, CS, RC | Validated | Dep, Anx internalizing problems | Children (GP and at-risk) | The Netherlands, UK, Italy, China, S. Korea, Switzerland, USA, Israel, Spain, Singapore, Canada, Argentina, Japan, Turkey, Germany | 2,5 | 5,5 |
| Carvalho 2022 [90] | Jun 2021 | 13 | 18,220 | CS | Validated self-report | Dep/Anx | Students | Kosovo, France, Turkey, Greece, Italy, Switzerland, Spain, Albania, Germany | 2 | 5 |
| Castaldelli Maia 2021 [30] | July 2020 | 58 | 193,137 | CS, RCT, L, CC | Self-report | Dep, Anx | GP, students, patients (mixed), HCW | China, Japan, Switzerland, Saudi Arabia, Serbia, Cyprus, Nepal, Brazil, Pakistan, UAE, Nigeria, Vietnam, Austria, Jordan, Spain, Albania, USA, Norway, India, Bangladesh, UK, Italy, Germany, Russia, Iran, Korea | 2,5 | 8,5 |
| Cenat 2021 [91] | May 2020 | 55 | 189.159 | CS | Validated | Dep, Anx, PTSD | HCW, GP | China, Italy, India, Singapore, France, USA, Iran, Vietnam, Spain, Turkey, Italy, Israel, Bolivia, Ecuador, Malaysia, Pakistan, Peru, and multiple country studies | 3 | 8,5 |
| Cenat 2022 [92] | Sep 2021 | 64 | 170,827 | L | Validated self-report structured clinical interviews | Dep, Anx, PTSD | Any population | North America (20), UK (7), and Italy (6). Four from: China, Spain; 3 from the Netherlands, Australia, Germany, Japan. Two from Argentina. One paper from: Estonia, Austria, Japan, France, Brazil, Colombia, Singapore, Ireland, and Sweden. | 4,5 | 11 |
| Cevik 2022 [144] | May 2021 | 48 | 77,616 | CS, L | Validated measures | Dep, Anx | Pregnant women | Israel, Turkey, Ethiopia, Bangladesh, Italy, China, Denmark, USA, S Africa, Ethiopia, Iran, ABD, Finland, Ghana, Vietnam, Croatia, Turkey, Malaysia, India, Pakistan, Poland | 2,5 | 8 |
| Chai 2021 [93] | March 2021 | 12 | 34,276 | CS | Validated | Dep, Anx | Children and adolescents | China | 2,5 | 7 |
| Chang 2021 [28] | Nov 2020 | 16 | 135 018 | CS, L | Validated measures | Dep, Anx | Students | France, Malaysia, Turkey, America, China, Poland, India, Bangladesh, Greece | 2,5 | 8 |

(*Continued*)

**Table 1.** (Continued)

| Study | End date search | Studies | Sample size | Study designs | Assessment | Outcomes[1] | Study populations | Countries/continents/ WHO regions | AMSTAR 2 scores | |
|---|---|---|---|---|---|---|---|---|---|---|
| | | | | | | | | | Critical | Total |
| Chekole and Abate 2021 [79] | Apr 2020 | 21 | 72,999 | CS, OBS | Self-report | Dep, Anx | GP, patients (Cov), students, women, HCW, children | China, India, Lebanon, Singapore, Mexico, USA, Spain, Iran, Jordan, Vietnam, Italy, UK, Ethiopia, Saudi | 4 | 11 |
| Chen 2021 [71] | Feb 2021 | 28 | 15,071 | CS, Cohort | Validated measures | Dep, Anx | HCW, gen pop, students | Cameroon, Egypt, Ethiopia, Libya, Mali, Morocco, Nigeria, RDC, Rwanda, S Africa, Togo, Tunisia | 2 | 7,5 |
| Chen 2022 [95] | May 2021 | 13 | 41.729 | CS | Validated measures | Dep | Children, adolescents | China | 2,5 | 7,5 |
| Chen 2022 [96] | Nov 2021 | 8 | 6,480 | CS | Validated measures | Dep, Anx | Postpartum women | Mexico, Myanmar, Turkey, UK, Ireland, Norway, Switzerland, the Netherlands, Italy, Canada | 1,5 | 6,5 |
| Ching 2021 [97] | March 2021 | 148 | 159,194 | CS | CS | Dep Anx | HCW | China, Turkey, Saudi Arabia, India, Pakistan, Indonesia, Nepal, Malaysia, Singapore, Japan, Iran, Oman, Jordan, Philippines, Bangladesh, Korea, Qatar, Iraq, Egypt | 4,5 | 10,5 |
| Dal Santo 2021 [98] | Aug 2021 | 12 | 48,344 | L | Validated measures | Dep, Anx | Any population | China, USA, Australia, Spain, UK, India, Switzerland, the Netherlands | 4,5 | 10,5 |
| da Silva 2021 [99] | May, 2021 | 7 | 7,102 | CS | Validated self-report | Dep Anxiety | Students | China | 1,5 | 4,5 |
| Demissie 2021 [31] | Sept 2020 | 19 | 18,335 | CS | Self-report | Dep, Anx, | Perinatal women | Colombia, Sri Lanka, Belgium, China, Canada, Iran, Turkey, Bosnia Herzegovina, Serbia, Ireland, UK, USA, Italy | 3 | 9,5 |
| Deng 2021 [33] | May 2020 | 34 | 29,996 | CS | NR | Dep, Anx | GP, HCW | China | 5 | 12,5 |
| Deng 2021 [32] | Jan 2021 | 89 | 1,441,828 | CS, L | Validated self-report | Dep, Anx | University Students | Italy, Turkey, Ethiopia, USA, France, China, Bangladesh, Spain, Switzerland, Ireland, Malaysia, Taiwan, South Korea, the Netherlands, Lebanon, UK, Slovakia, Egypt, Russia, Belarus, Saudi Arabia, Jordan, India, Ukraine, Poland, UAE, Pakistan, Argentina | 4,5 | 11 |
| de Sousa 2021 [100] | March 2021 | 18 | NR | MA | NR | Dep, Anx, PTSD | GP, HCW | Asia, Europe, South America, Central America, North America, Oceania | 3,5 | 7 |
| Delanerolle 2022 [101] | Aug 2021 | 188 | NR | PC | NR | Dep, Anx, PTSD | HCW, GP, Patients | NR | 4,5 | 8 |
| Dong 2021 [34] | Oct 2020 | 22 | NR | CS, L | Validated measures | Dep, Anx, PTSS | HCW | China | 2,5 | 9,5 |
| Dong 2021 [35] | Oct 2020 | 38 | NR | CS, L | | Dep, Anx, PTSS | Patients (Cov) | China, Italy, Iran, India, Korea, Ecuador, Switzerland, Germany | 2,5 | 10 |

*(Continued)*

**Table 1.** (Continued)

| Study | End date search | Studies | Sample size | Study designs | Assessment | Outcomes[1] | Study populations | Countries/continents/ WHO regions | AMSTAR 2 scores | |
|---|---|---|---|---|---|---|---|---|---|---|
| | | | | | | | | | Critical | Total |
| Dragioti 2022 [102] | Sep 2020 | 173 | 502,261 | OBS | NR | Dep, Anx, PTSD | HCW, GP, patients (COVID and other), students, caregivers/ family | China, Italy, India, USA, Australia, Brazil, Canada, Egypt, Turkey, Iran, Japan, S Korea, UK, Ireland, Spain, France, Germany, Poland, Sweden, Croatia, Greece, Cyprus, Jordan | 6 | 12 |
| Dutta 2021 [36] | Aug 2020 | 33 | 39,703 | CS | Validated measures | Dep, Anx, | HCW | Singapore, India, China, Turkey, Brazil, Italy, Poland, Pakistan, Iran, Jordan, Nepal, USA | 3 | 9,5 |
| Ebrahim 2022 [103] | Sep 2020 | 90 | 46,284 | Quantitative | Validated | Dep, Anx, PTSD | University students | USA, India, Turkey, Israel, Iran, Jordan, Australia, Russia, China, KSA, Egypt, Poland, Brazil, Canada, Pakistan, Philippines, Morocco, Italy, Albania | | |
| El-Qushayri 2021 [37] | Jan 2021 | 8 | 3,137 | CS | Validated measures | Dep, Anx | HCW | Egypt | 2,5 | 7 |
| Fan 2021 [38] | Oct 2020 | 158 | 515,452 | CS, SR | Validated self-report | Dep, Anx, PTSS | GP, HCW, patients (COVID) | China, India, Spain, Greece, Turkey | 1,5 | 3,5 |
| Fang 2022 [104] | March 2022 | 104 | 2,088,032 | CS | Validated self-report | Dep, Anx, stress | Students | China, Korea, Malaysia, Italy, Ethiopia, America, Asia, Australia, Palestine, Saudi Arabia, India, Lithuania, Poland, Germany, Bhutan, Bengal, Spain, Brazil, Uganda, Nigeria, Thailand, Japan, Mexico, Switzerland, Czech | 6 | 12 |
| Ghahramani 2022 [143] | February 2022 | 44 | NR | CS, CC | Validated self-report | Dep, Anx, PTSD, stress | HCW | China, Italy, USA, Oman, India, Iran, Turkey, Pakistan, Israel, Singapore, Russia, Nepal, global, South Korea, Jordan, Iraq, Japan | 4 | 10 |
| Ghazanfarpour 2021 [39] | NR | 11 | NR | CS, OBS | Validated/ unvalidated self-report | Dep, Anx | Pregnant women | Belgium, Greece, Iran, Pakistan, Canada, Italy, Sri Lanka, China, Turkey | 3,5 | 7,5 |
| Guo 2021 [40] | Jul 2020 | 11 | 25,020 | CS | Validated self-report | Dep (levels) | Students | China | 2,5 | 7 |
| Halemani 2021 [105] | April 2021 | 13 | 90,601 | CS | Validated self-report | Dep, Anx, stress | HCW (doctors, nurses) | China, Singapore, India, Nepal, Turkey, Japan, UK, Saudi Arabia | 5 | 10 |
| Hao 2021 [41] | Apr 2021 | 20 | 10,886 | CS | Validated measures | Dep, Anx, OCD, phobia | HCW | China, Singapore | 4 | 10,5 |
| Hosen 2021 [106] | March 2021 | 24 | 49,806 | CS | Validated self-report | Dep, Anx, stress | Students, GP (incl COVID-19 patients, quarantined people), HCW | Bangladesh | 3 | 8 |
| Hossain 2021 [42] | Oct 2020 | 35 | 41,402 | CS | Validated self-report | Dep, Anx | GP, HCW | India, Bangladesh, Pakistan, Nepal, Sri Lanka | 2,5 | 8 |
| Hu 2022 [107] | March 2021 | 71 | 98,533 | Empirical studies | Validated self-report | Dep, Anx | HCW | China | 5 | 10 |

*(Continued)*

**Table 1.** (Continued)

| Study | End date search | Studies | Sample size | Study designs | Assessment | Outcomes[1] | Study populations | Countries/continents/ WHO regions | AMSTAR 2 scores | |
|---|---|---|---|---|---|---|---|---|---|---|
| | | | | | | | | | Critical | Total |
| Huang 2022 [108] | 2022 | 17 | 8,096 | CS, PC | Validated self-report | Dep, Anx, stress | HCW | Qatar, Peru, Germany, India, USA, Ecuador, Australia, Sri Lanka, Pakistan, Nepal, Germany, Ethiopia, China | 6 | 12 |
| Jia 2022 [109] | August 2021 | 41 | 36,608 | CS, LS | Validated self-report | Dep, Anx | (Medical) students | Nepal, Jordan, Turkey, Libya, China, America, India, Brazil, Germany, Pakistan, Iran, Japan, Greece, Spain, Albania, France, Bangladesh | 5 | 12 |
| Johns 2021 [110] | March 2021 | 33 | 31,447 (Dep); 33,281 (Anx) | CS | Validated self-report | Dep, Anx | HCW (doctors) | Cyprus, Brazil, USA, Turkey, Libya, Malaysia, France, UK, South America, India, Pakistan, China, Columbia, Germany, Croatia, global | 5 | 12 |
| Kan 2021 [111] | Feb 2021 | 103 | 140,732 | CS, CC, L, cohort | NR | Anx | General public, COVID-19 patients | Continents: Africa, America, Europe, Asia/ WHO regions: AFRO, EMRO, SEARO, EURO, PAHO, WPRO | 3,5 | 8 |
| Khraisat 2022 [112] | August 2021 | 13 | 3,056 | CS, L | Validated self-report, DSM-5 criteria | Dep, Anx | Patients with eating disorders | Germany, Australia, Spain, USA, the Netherlands, Canada, UK, Italy, Sweden | 4 | 8 |
| Knox 2022 [113] | March 2021 | 33 | 131,844 | CS, L | Validated self-report | Dep, Anx, stress | GP (under social restrictions) | Italy, Germany, Brazil, USA, Switzerland, Greece, UK, Norway, China, Argentina, Australia, Spain, New Zealand, Hong Kong, international | 4 | 9 |
| Kunzler 2021 [43] | May 2020 | 43 | 71,613 | CS, L | Validated self-report | Dep, Anx | GP, HCW, students, | Iraq, UK, Germany, Italy, Spain, Croatia, Iran, USA, Turkey, Taiwan, Hong Kong, Macao, Russia, Belarus, India, Bangladesh, Italy, Greece, France, Oman, international, Canada, Saudi Arabia, Pakistan, Singapore, Jordan, Israel | 2,5 | 10 |
| Kuroda 2021 [114] | March 2021 | 28 | 7,959 | CS, cohort, case | Validated self-report | Dep, Anx | Patients (epilepsy) | Kuwait, Spain, Saudi Arabia, Italy, China, Malaysia, USA, Lithuania, India, UK, Iran, Brazil, Belgium, the Netherlands, Turkey, Australia, international | 3 | 8 |

(*Continued*)

**Table 1.** (*Continued*)

| Study | End date search | Studies | Sample size | Study designs | Assessment | Outcomes[1] | Study populations | Countries/continents/ WHO regions | AMSTAR 2 scores | |
|---|---|---|---|---|---|---|---|---|---|---|
| | | | | | | | | | Critical | Total |
| Lee 2021 [44] | Sept 2020 | 114 | 640,037 | NR | Validated self-report | Dep | GP | USA, Vietnam, China, Italy, UK, Albania, Austria, Bangladesh, Bosnia and Herzegovina, Brazil, Canada, Ecuador, France, Greece, Germany, Hong Kong, India, Iran, Ireland, Israel, Japan, Jordan, Kenya, Mexico, Nepal, Norway, Pakistan, Poland, Saudi Arabia, South Korea, Spain, Sweden, Switzerland, Turkey, USA | 4 | 6 |
| Lee 2022 [115] | April 2021 | 6 | 3,127 | CS | Validated self-report, not reported | Dep, Anx | Patients (HIV) | USA, Argentina, Italy, Kenya, Turkey, India, Belgium | 3 | 7 |
| Li 2021 [47] | Aug 2020 | 65 | 97,333 | NR | Validated self-report | Dep, Anx, PTSS | HCW | Italy, Thailand, China, Spain, Oman, India, UK, Singapore, Hong Kong, Italy, Argentina, Brazil, Mexico, Chile, Togo, Turkey, USA, Jordan, Iran, Pakistan, Taiwan, Switzerland, Saudi Arabia | 4 | 9 |
| Li 2021 [45] | Dec 2020 | 66 | 599,679 | CS | Validated self-report | Dep, Anx | GP | China | 3,5 | 10,5 |
| Li 2021 [46] | Oct 2020 | 27 | 706,415 | CS | Validated self-report | Dep, Anx | Students | China, France, USA, Jordan, South Korea, Japan, Spain, Bangladesh, Lebanon, Switzerland, Israel | 4,5 | 9,5 |
| Liyanage 2022 [116] | Feb 2021 | 36 | NR | CS | Validated self-report | Anx | University students | China, Bangladesh, Malaysia, Turkey, India, Nepal, Saudi Arabia, Jordan, USA, Egypt | 2,5 | 8 |
| Liu 2021 [48] | Jul 2020 | 71 | 146,139 | CS, L | Validated self-report | Dep, Anx, PTSS | GP, patients (COVID-19) | China, Italy, Turkey, Spain, Greece, India, Singapore, USA | 3,5 | 10 |
| Liu 2021 [49] | Apr 2021 | 21 | 38,372 | NR | Validated self-report | Dep, Anx | HCW | India, USA, China, Turkey | 2 | 5 |
| Liu 2021 [48] | Dec 2020 | 22 | 4,318 | NR | Validated self-report | Dep, Anx, | Patients (COVID-19) | China, South Korea, India, Ecuador, Jordan, Turkey, Italy, Iran | 4,5 | 10 |
| Luo 2021 [51] | Feb 2021 | 84 | 1,292,811 | CS | Validated self-report | Dep, severe Dep | Students | South Korea, China | 5 | 13 |
| Ma 2021 [52] | Sep 2020 | 23 | 46 to 9,554 | CS, L | Validated measures | Dep, Anx, PTSS | Children, adolescents | Turkey, China | 3,5 | 9,5 |
| Ma 2022 [117] | July 2021 | 54 | 256,896 | CS, L | Validated/ unvalidated measures | Dep, Anx | Teachers | China, Italy, USA, Spain, Turkey, Canada, Chile, Australia, Ecuador, Brazil, India, Israel, Greece, Germany, Japan, Jordan, Mexico, Pakistan, Philippines, Portugal, Saudi Arabia, Slovakia, UK | 2,5 | 8 |

(*Continued*)

**Table 1.** (Continued)

| Study | End date search | Studies | Sample size | Study designs | Assessment | Outcomes[1] | Study populations | Countries/continents/WHO regions | AMSTAR 2 scores | |
|---|---|---|---|---|---|---|---|---|---|---|
| | | | | | | | | | Critical | Total |
| Mahmud 2021 [53] | Sep 2020 | 83 | 160,477 | CS | Validated measures | Dep, Anx | HCW | China, Singapore, India, Lebanon, Greece, Bangladesh, Philippines, Nepal, Egypt, Oman, Turkey, Canada, USA, Poland, Spain, Pakistan, Italy, Jordan, Korea, South Korea, Saudi Arabia, UK, Vietnam, Finland, Australia, Ghana, Iran, Croatia, Germany | 3 | 10 |
| Makwana 2022 [118] | Mar 2022 | 6 | 3,248 | CS | Validated measures | Dep | Medical students | India | 1 | 5,5 |
| Marvaldi 2021 [78] | Oct 2020 | 70 | 101,017 | CS | Validated measures | Dep, Anx | HCW | Iraqi Kurdistan, Saudi Arabia, Thailand, Egypt, France, Turkey, India, Ireland, Italy, China, Singapore, Spain, Pakistan, Bahrain, Nepal, USA, Philippines, Iran, Oman, Germany | 4 | 10,5 |
| Mulyadi 2021 [119] | Jun 2021 | 17 | 13,247 | CC, CS, Cohort | Validated/unvalidated measures | Dep, Anx, PTSS | Nurses | China, Turkey, India, Nepal, USA, Australia, Indonesia, Israel | 2,5 | 8,5 |
| Nagarajan 2022 [120] | May 2021 | 13 | 1,093 | Obs, CS | Validated measures | PTSS | Severe COVID-19 patients | Italy, the Netherlands, UK, France, Turkey, China, Iran, USA | 3,5 | 9 |
| Natarajan 2022 [121] | Jun 2021 | 36 | 11,598 | Cohort, CS | NR | Dep, Anx | Long COVID patients | NR | 3,5 | 9,5 |
| Necho 2021 [54] | Nov 2020 | 16 | 78,225 | CS, L | Validated self-report | Dep, Anx, PTSS | GP | China, Italy, Australia, Turkey, France, India, Iran | 2 | 5 |
| Nochaiwong 2021 [55] | Jul 2020 | 107 | 398,771 | CS, L | Validated self-report | Dep, Anx, PTSS | GP | Nigeria, South Africa, Brazil, Mexico, USA, Bangladesh, India, Nepal, Thailand, Germany, Greece, Ireland, Italy, Norway, Portugal, Spain, Sweden, Turkey, UK, Egypt, Iran, Jordan, Pakistan, Saudi Arabia, Tunisia, United Arab Emirates, Australia, China (including Hong Kong, Macau, Taiwan), Japan, Malaysia, New Zealand, Vietnam | 5 | 10,5 |
| Norhayati 2021 [56] | Apr 2021 | 80 | 149,925 | CS, CC, L | Validated self-report | Dep, Anx, PTSS | HCW | Western Asia, Southern Asia, Eastern Asia, South-Eastern Asia | 2,5 | 5 |
| Olaya 2021 [57] | Sept 2020 | 57 | 46 to 14,825 | CS | Validated self-report | Dep | HCW (3 groups) | USA, China, Italy, Russia, Spain, Japan, Mexico, The Republic of Korea, Turkey, Malaysia, Singapore, India, Canada, Egypt, Australia, Portugal, Libya, international | 1,5 | 6 |

(*Continued*)

**Table 1.** (Continued)

| Study | End date search | Studies | Sample size | Study designs | Assessment | Outcomes[1] | Study populations | Countries/continents/ WHO regions | AMSTAR 2 scores | |
|---|---|---|---|---|---|---|---|---|---|---|
| | | | | | | | | | Critical | Total |
| Ozamiz-Etxebarria 2021 [58] | June 2021 | 8 | 93,886 | CS | Validated self-report | Dep, Anx | Teachers | International, Jordan, Brazil, USA, India, China, Spain | 2,5 | 7 |
| Panda 2021 [122] | August 2020 | 15 | 22,996 | CS | Validated/ unvalidated | Dep, Anx | Children, caregivers | France, Italy, China, Spain, India, Hong Kong, Brazil, Turkey, Bangladesh, Korea | 3,5 | 7,5 |
| Pappa 2022 [123] | Feb 2021 | 25 | 20,352 | CS, cohort | Validated self-report | Dep, Anx | General public, Frontline HCW, General HCW, students | Indonesia, Malaysia, Philippines, Singapore, Thailand, Vietnam | 5 | 12,5 |
| Phiri 2021 [9] | Jan 2021 | 206 | NR | CS, L | Validated self-report | Dep, Anx, PTSS | GP, HCW | International, China, Singapore, India, Spain, Turkey, Italy, Germany, Iran, Bangladesh, USA, Pakistan, Denmark, UK, Australia, Egypt, Jordan, Malaysia, Poland, Portugal, New Zealand, Ireland, Brazil, Switzerland, Norway, Oman, Saudi Arabia, UAE, Iraq, Canada, Austria, Argentina, Chile, Sweden, Philippines, UK, Vietnam, Colombia, Hong Kong, Morocco, France, Russia, Taiwan, Japan, Georgia, Tunisia, South Korea, Indonesia, Peru, Paraguay, DCR, Ethiopia, the Netherlands, Belgium, Israel, Togo, Rwanda, Haiti, Greece, Palestine, Iran, Czech Republic, Nepal, Serbia, Mexico | 3,5 | 9 |
| Prati 2021 [59] | Jun 2020 | 25 | 72,004 | CS, L, CT | Unspecified | Dep, Anx, PTSS | GP | Europe, North America, Asia, Oceania | 1,5 | 7 |
| Premraj 2022 [124] | Aug 2021 | 18 | 10,530 | CS, CC, L, cohort | NR | Dep, Anx | COVID-19 patients | NR | 3,5 | 7,5 |
| Qi 2022 [125] | NR | 28 | 20,891 | CS | Validated self-report | PTSD | HCW | Saudi Arabia, Ethiopia, France, Turkey, China, Italy, Canada, the Netherlands, USA, Korea, UK | 4 | 10 |
| Qiu 2021 [60] | Apr 2020 | 27 | 34 842 | CS, L, CC | Validated self-report | PTSS | GP, HCW | China, India, Singapore, Greece, Ireland unreported | 3 | 9,5 |
| Qiu 2021 [60] | Oct 2020 | 106 | NR | CS, L | Validated self-report | PTSS | GP, HCW, patients | China, Singapore, Japan, Canada, Hong Kong, Taiwan, Greece, South Korea, Brazil, Mexico, International, Australia, Italy, Japan, Spain, Tunisia, Egypt, USA, Ireland, Israel, France, Vietnam, Germany, Austria, Saudi Arabia, India, Philippines | 3 | 8,5 |

(*Continued*)

**Table 1.** (Continued)

| Study | End date search | Studies | Sample size | Study designs | Assessment | Outcomes[1] | Study populations | Countries/continents/ WHO regions | AMSTAR 2 scores | |
|---|---|---|---|---|---|---|---|---|---|---|
| | | | | | | | | | Critical | Total |
| Racine 2021 [63] | Feb 2021 | 29 | 80,879 | NR | Validated self-report | Dep | Children, adolescents | China, USA, Jordan, Ecuador, Italy, Spain, Portugal, Brazil, Greece, Canada, Germany | 2 | 8 |
| Racine 2021 [126] | March 2021 | 18 | 8,987 | NR | Validated self-report | Dep, Anx | Mothers of young children | Europe, East Asia, North America, the Middle East, South Asia, Southeast Asia, South America | 5 | 11,5 |
| Rezaei-Hachesu 2022 [127] | June 2021 | 10 | 4,816 | NR | Validated self-report | Dep, Anx | HCW | Iran | 1 | 7 |
| Robinson 2021 [10] | Jan 2021 | 61 | 55,015 | L | Validated self-report | Dep, Anx, MHS, PTSS | GP, HCW | Europe, North America, China | 5,5 | 10 |
| Raoofi 2021 [62] | Feb 2021 | 46 | 61,551 | CS, L | | Anx | HCW | North America, South America, Europe, Africa, Southeast Asia, Asia, Eastern Mediterranean | 2 | 7 |
| Shorey 2021 [68] | Dec 2020 | 26 | 24,040 | CS, CC, Mixed | Self-report | Dep, Anx | Ante-, peri-, and postnatal women | Canada, Belgium, Greece, Turkey, China, Iran, USA, Hong Kong, Italy, Japan, Israel, Sri Lanka | 3,5 | 8,5 |
| Salehi 2021 [64] | May 2020 | 13 | 11,217 | CS, CC | Validated self-report | PTSS | GP | Canada, Singapore, China, India, Spain, South Korea, Taiwan, Hong Kong | 3,5 | 8 |
| Santabárbara 2021 [65] | Sep 2020 | 71 | 46 to 8,817 | CS | Validated self-report | Anx | HCW | Thailand, Turkey, Oman, China, Ecuador, India, Singapore, Italy, Spain, Libya, Kosovo, Nepal, USA, Cameroon, Jordan, Croatia, Germany, Serbia, Saudi Arabia, Poland, South Korea, Bolivia, Peru | 4 | 8 |
| Santabarbara 2021 [128] | Aug 2021 | 15 | 6,141 | CS | Validated self-report | Anx | Dental students | USA, Peru, Malaysia, Brazil, Saudi Arabia, Turkey, Italy, Palestine, Germany, UAE | 3,5 | 8,5 |
| Santabarbara 2021 [130] | Dec 2020 | 11 | 6,576 | CS | Validated self-report | Dep | Medical students | Kazakhstan, Libya, Morocco, China, Iran, India, Japan, Brazil | 3,5 | 7,5 |
| Santabarbara 2021 [129] | Aug 2021 | 13 | 4.147 | CS | Validated self-report | Dep | Dental students | Iran, India, USA, Turkey, Saudi Arabia, Palestine, Brazil, Germany, Malaysia | 4 | 9 |
| Santomauro 2021 [5] | Jan 2021 | 48 | NR | L, CS | Validated screening measures | Dep, Anx | General population | China, Australia, USA, New Zealand, Japan, Norway, UK, Ireland, Germany, the Netherlands, France, Spain, Austria, Denmark, Czech Republic | 4 | 10,5 |
| Safi-Keykalah [131] | Aug 2021 | 24 | 13.169 | CS, RC, CC | Validated self-report | Dep (postpartum) | Perinatal women | China, Italy, UK, Saudi Arabia, Ireland, Norway, Switzerland, the Netherlands, Serbia, Turkey, Japan, Brazil, Spain, Israel, Belgium, Hong Kong, Argentina, Mexico, Poland | 2,5 | 8 |

(*Continued*)

**Table 1.** (Continued)

| Study | End date search | Studies | Sample size | Study designs | Assessment | Outcomes[1] | Study populations | Countries/continents/ WHO regions | AMSTAR 2 scores | |
|---|---|---|---|---|---|---|---|---|---|---|
| | | | | | | | | | Critical | Total |
| Saragih 2021 [66] | Nov 2020 | 38 | 53,784 | CS, CC | Validated instrument | Dep, Anx, PTSS | HCW | China, Italy, India, USA, Australia, Nepal, Iran, Saudi Arabia, Canada, Egypt, France, Mali, Norway, Poland, Oman, Serbia, Spain, South Korea, and Turkey | 1,5 | 5,5 |
| Schafer 2022 [132] | July 31 2020 | 36 | NR | CS, L | Self-report | Anx, Dep | GP, HCW | Americas, Eastern Mediterranean, Europe, Pacific | 1 | 2 |
| Sharma 2022 [133] | Oct 2021 | 22 | 16,424 | CS, L | Validated self-report | Anx | GP | India | 4 | 8 |
| Sideli 2021 [69] | Jan 2021 | 26 | 3,399 | CS, L, CC, | Validated/ unvalidated measures | Dep, Anx | Eating disorder patients/obesity | Spain, Italy, Portugal, UK, USA, Ireland, Australia, Germany, Canada, the Netherlands | 3,5 | 8,5 |
| Singh 2021 [67] | Oct 2020 | 22 | 9,947 | CS | Validated self-report | Dep | GP, HCW | India | 3 | 4,5 |
| Ślusarska 2022 [134] | Feb 2021 | 23 | 44,165 | CS, L (1) | Validated self-report | Dep, Anx | HCW (nurses) | China, Philippines, USA, Turkey, Saudi Arabia, Iran, Great Britain, Brazil, and Canada | 4 | 10 |
| Sun 2021 [80] | Sep 2020 | 47 | 81,277 | CS | Validated measures | Dep, Anx | HCW | Iran, China, Singapore, France, Ecuador, Libya, Italy, Philippines, Jordan, Pakistan, Poland, Brazil, America | 1,5 | 6,5 |
| Tomfohr-Madsen 2021 [70] | Feb 2021 | 46 | NR | CS, OBS | Validated self-report/clinical interview | Dep, Anx | Antenatal women | International, Iran, China, Spain, Greece, Canada, Ireland, the Netherlands, Switzerland, Norway, Belgium, Turkey, Qatar, USA, UK, Japan, Argentina, Italy, Mexico, Singapore, Sri Lanka, Pakistan, Poland | 3,5 | 8 |
| Varghese 2021 [73] | Oct 2020 | 27 | NR | CS | | Dep, Anx, PTSS | HCW | Germany, Croatia, Poland, Russia, Italy, Jordan, China, Vietnam, Turkey, Singapore, Philippines, Oman, Iran, India | 3,5 | 9 |
| Wang 2021 [71] | Sep 2020 | 28 | 436,799 | CS | Validated self-report | Dep, Anx | College students | China, non-China | 2,5 | 5 |
| Xie 2021 [72] | Mar 2021 | 12 | 1,705 | CS, OBS, RCT | Validated self-report | Dep, Anx | COVID-19 patients | China | 2,5 | 8 |
| Xiong 2022 [135] | June 2020 | 44 | 65,706 | CS, interventions | Validated self-report | Anx, Dep, PTSS | HCW | China | 4,5 | 10,5 |
| Yan 2021 [136] | Sep 2020 | 28 | 436,799 | CS, L | Validated self-report | Dep, Anx | HCW | China, Hong Kong | 4 | 9 |
| Yan 2022 [136] | March 2021 | 17 | 11,237 | OBS | Validated tools | Anx, Dep | Older adults (COVID-19, GP, chronic disease) | Mainland China, Hong Kong | 4 | 9 |

*(Continued)*

**Table 1.** (Continued)

| Study | End date search | Studies | Sample size | Study designs | Assessment | Outcomes[1] | Study populations | Countries/continents/ WHO regions | AMSTAR 2 scores | |
|---|---|---|---|---|---|---|---|---|---|---|
| | | | | | | | | | Critical | Total |
| Yang 2022 [137] | May 2021 | 10 | 17,385 | CS, case OBS | Validated tools/clinical diagnosis | PTSD | Children | China, USA, Italy | 2,5 | 8 |
| Yunitri 2022 [138] | June 2021 | 63 | 124,952 | CS, L | Validated self-report | PTSD | Patients (cov), HCW, GP | China (24), Singapore, India, Malaysia, Indonesia, South Korea, Vietnam, Europe (i.e., France, Greece, Italy, Norway, Spain, Ireland, Poland), Canada, USA, Tunisia, Saudi Arabia, Brazil, Mexico, Israel, Turkey | 4,5 | 9,5 |
| Zhang 2021 [75] | May 2020 | 26 | 22,062 | CS | Validated self-report | Dep, Anx | HCW | China | 4 | 9,5 |
| Zhang 2021 [76] | Aug 2020 | 11 | NR | CS | Self-report | PTSD | GP | China, Italy, Spain, Israel, USA, Ireland | 4 | 10,5 |
| Zhang, Chen 2022 [139] | Feb 2021 | 28 | 86,323 | CS, L | Validated self-report | Dep, Anx | GP, HCW, students | Spain | 3 | 9 |
| Zhang 2022 [140] | Aug 2021 | 62 | 196,950 | CS, L | Validated self-report | Dep, Anx | GP, HCW (general and frontline), students | Latin American countries (Argentina, Bolivia, Brazil, Chile, Colombia, Ecuador, Haiti, Mexico, Panama, Paraguay, Peru, Trinidad and Tobago, mixed) | 3 | 9 |
| Zhang 2022 [141] | Feb 2021 | 21 | NR | CS, L | Validated self-report | Dep, Anx | GP, HCW (general and frontline), students | Eastern Europe and Russia (Albania, Armenia, Azerbaijan, Belarus, Bosnia and Herzegovina, Bulgaria, Croatia, Czech Republic, Georgia, Hungary, Kosovo, Moldova, Montenegro, North Macedonia, Poland, Romania, Russia, Serbia, Slovakia, Slovenia, Turkey, and Ukraine). | 3 | 9 |
| Zhao 2021 [77] | May 2020 | 36 | NR | CS | Validated self-report | Dep, Anx, PTSS | GP | China, Hong Kong, Vietnam, Israel, Spain, Italy, Taiwan, Singapore, India, Canada | 4,5 | 12 |
| Zhu 2021 [142] | May 2021 | 176 | 1,732,456 | CS, L | Validated self-report | Dep, Anx | Students | Countries from East Asia, Europe, South Asia, Middle East, North America, Southeast Asia, Africa (4), Central America (3), Oceania (1), multiple geographical regions (2) | 4,5 | 11 |

[1] Selected common mental disorders outcomes (depression, anxiety and PTSD).

CC, case-control; CS, cross-sectional; L, longitudinal; NR, not reported; OBS, observational; PC, prospective cohort; RC, retrospective cohort; RCT, randomized controlled trial.

Robinson and colleagues [10] with changes estimated from longitudinal data and the reviews of Fang and colleagues [104] and Huang and colleagues [108] with pooled prevalence rates based on cross-sectional data, fulfilled all critical AMSTAR 2 ratings.

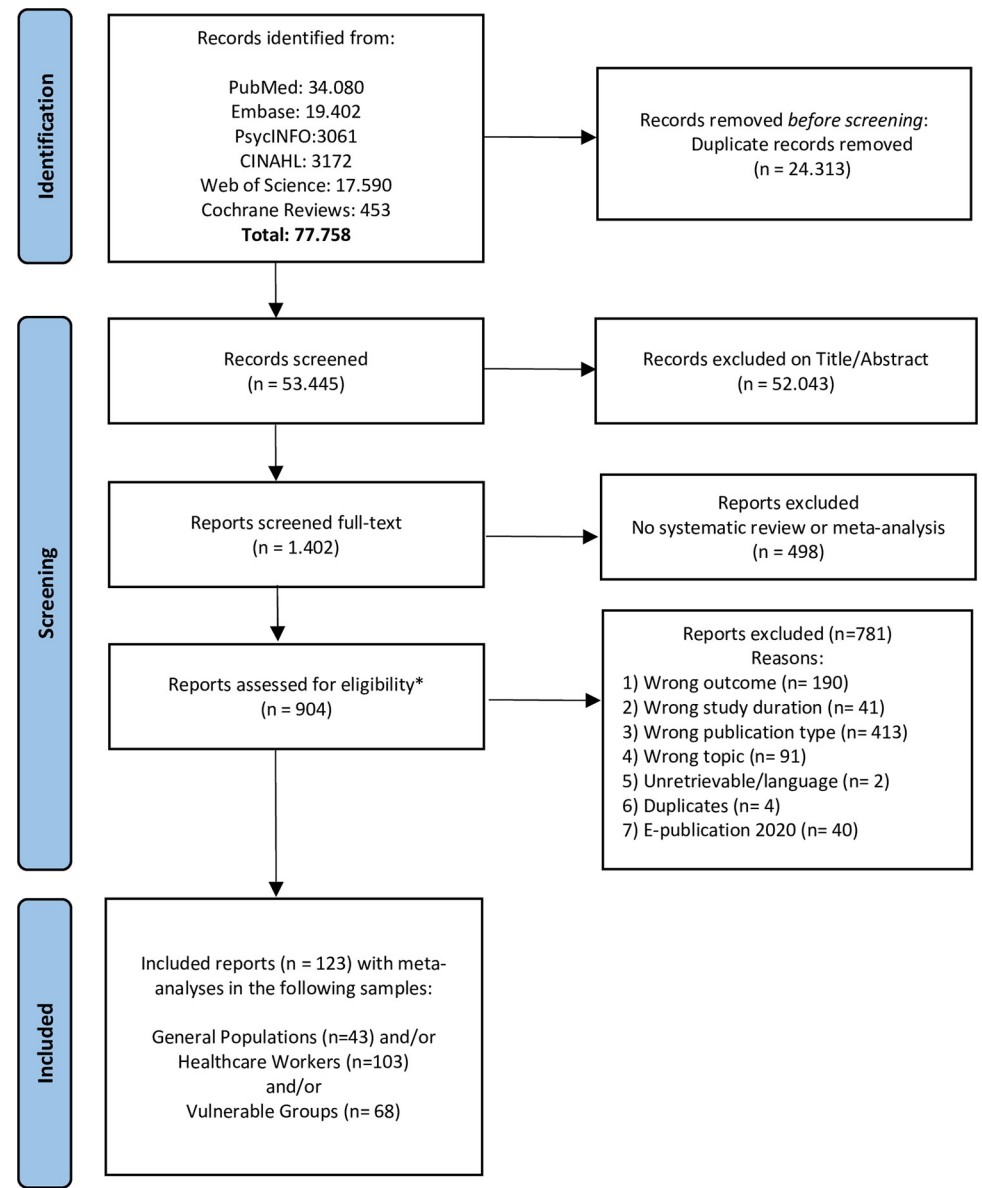

**Fig 1. PRISMA flow chart of study selection based on both the initial and updated search.** Caption credit: Page and colleagues [21]. The PRISMA 2020 statement: an updated guideline for reporting systematic reviews. BMJ 2021;372: n71. https://doi-org.vu-nl.idm.oclc.org/10.1136/bmj.n71.

## Association between COVID-19 and symptoms of common mental disorders

**General (or mixed) population.** Reviews with meta-analyses of longitudinal (within-person) data showed that symptoms of depression or mood disorder were increased during- compared to pre-pandemic periods (SMC: 0.22, 95% CI: 0.13 to 0.30 [10]; Hedges' g: 0.16, 95% CI: 0.01 to 0.30 [59]; increase of 27.6%, 95% CI: 25.1 to 30.3 [5]) and remained increased over time in the first half year of 2020 (March to April SMC: 0.23, 95% CI: 0.11 to 0.34 and May to July SMC: 0.20, 95% CI: 0.10 to 0.30) [10] (**Table 2**). During social restrictions, depression symptoms were higher than in pre-implementation of public health and social measures or pre-pandemic periods (Cohen's d: 0.83, 95% CI: 0.30 to 1.37 [113]) and during pandemic

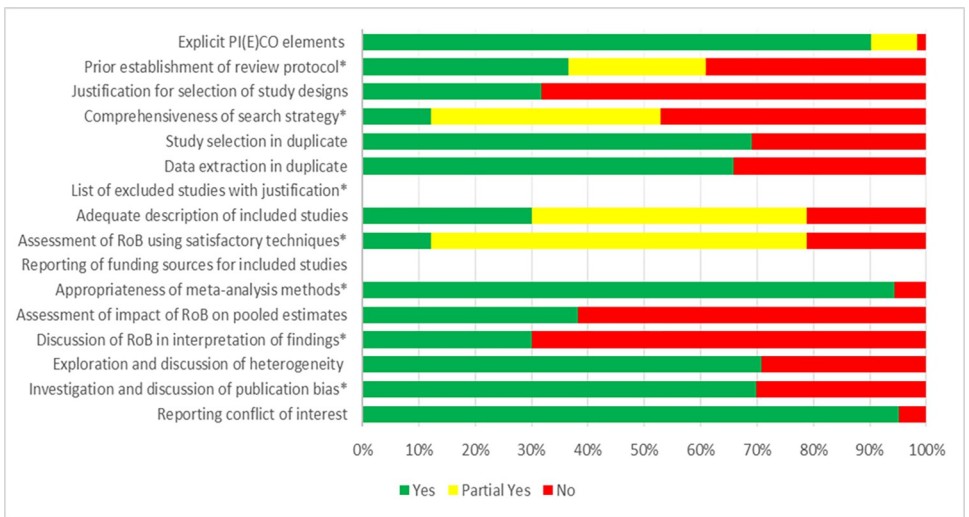

Fig 2. Quality assessments by outcome as percentages across all included systematic reviews with meta-analyses. P (I)ECO, Population, (Intervention) or Exposure, Comparator, Outcome; RoB, Risk of Bias.

depression symptoms compared to matched pre-pandemic cross-sectional data were significantly increased as well (standardized mean difference (SMD): 0.67, 95% CI: 0.07 to 1.27 [43]). Anxiety disorder symptoms were also higher during- compared to pre-pandemic periods (SMC 0.12, 95% CI: 0.02 to 0.23 [10]; 25.6% increase, 95% CI: 23.2 to 28.0 [5]) as were anxiety and posttraumatic stress symptoms when pooled together (Hedges' g: 0.18, 95% CI: 0.07 to 0.27) [59]. Anxiety symptoms during pandemic were also increased compared to pre-pandemic data (SMD: 0.40, 95% CI: 0.15 to 0.65) [43]. The effect size of social restrictions on anxiety symptoms was however not significant (Cohen's d: 0.26, −0.04 to 0.56) [113] and not higher in March to April 2020 (SMC: 0.14, 95% CI: −0.02 to 0.30) nor in May to July 2020 (SMC: 0.05, 95% CI: −0.04 to 0.14) [10]. For general mental health symptoms, significant differences during- versus pre-pandemic periods (SMC 0.11, 95% CI: 0.04 to 0.17 and SMC 0.17, 95% CI: 0.07 to 0.26) [59] and during social restrictions versus pre-implementation of public health and social measures or pre-pandemic (Cohen's d: 0.41, 95% CI: 0.17 to 0.65) [113] were found. High heterogeneity across studies was found in all reviews ($I^2 > 94\%$). Moderation analyses showed a significantly larger increase in depression than anxiety symptoms during the pandemic [10]. Pooled estimates of differences in change from during- to pre-pandemic by sex or gender showed that females had a significantly greater during pandemic increase in anxiety symptoms (SMD 0.15; 95% CI: 0.07 to 0.22) and in general mental health than males (SMD: 0.15, 95% CI: 0.12 to 0.18) [98]. Females worsened more during the pandemic in terms of anxiety and depression symptoms, as well as younger compared to older age groups [5]. The largest changes in prevalence rates of depression and anxiety symptoms were found in studies with data from early stages of pandemic [10], or when prevalence rates were compared with studies of older pre-pandemic data [43,10]. Changes in PTSD symptom levels could not be adequately assessed due to a lack of aftermath macro-stressors similar to the COVID-19 situation [43]. Country-level COVID-19 exposure factors (e.g., death rate, stringency measures) and individual-level factors (e.g., age, sex or gender) could not significantly explain heterogeneity in changes of depression and anxiety symptoms across studies in 2 reviews [10,59], while in other reviews, depression symptoms were higher in people exposed to strict compared to moderate restrictions [113] and anxiety and depression prevalence increased when human

**Table 2. Outcomes from meta-analyses on during- versus pre-pandemic longitudinal or comparative data.**

| Variables | Population | Studies (n) | Designs of included studies | Pooled sample size | Metric | Pooled effect | 95% CI change/ increase |
|---|---|---|---|---|---|---|---|
| **Depression** | | | | | | | |
| During- vs. pre-pandemic [10] | Mixed | 58 | Longitudinal (within person) | | SMC | **0.22** | **0.13 to 0.30** |
| March–April 2020 vs. pre-pandemic | Mixed | 58 | Longitudinal (within person) | | SMC | **0.23** | **0.11 to 0.34** |
| May–July 2020 vs. pre-pandemic | Mixed | 58 | Longitudinal (within person) | | SMC | **0.20** | **0.10 to 0.30** |
| During- vs. pre-pandemic [5] | General | 57 | Longitudinal (within person) or cross-sectional (if pre-pandemic available) | | % increase | **27.6** | **25.1 to 30.3** |
| During- vs. pre-pandemic [59] | General | 9 | Longitudinal (within-person), experimental (restrictions vs. no restrictions) | | Hedges' g | **0.16** | **0.01 to 0.30** |
| During- vs. pre-pandemic [98] | Mixed | 4 | Longitudinal (>90% within-person) | 4,475 | SMD | 0.12[1] | −0.09 to 0.33 |
| | Mixed | 1 | Longitudinal (>90% within-person) | 139 | PCD | 0.12[1] | −0.03 to 0.28 |
| Social restrictions vs. pre-pandemic/pre-PHSM [113] | General | 27 | Longitudinal, cross-sectional | | Cohen's d | **0.83** | **0.30 to 1.37** |
| During- vs. pre-pandemic [43] | General | 25 | Cross-sectional-observational (p) (c) | 60,213 (p) 183,747 (c) | SMD | **0.67** | **0.07 to 1.27** |
| During- vs. pre-pandemic [43] | HCW | 14 | Cross-sectional-observational (p) (c) | 2,226 (p) 4,605 (c) | SMD | -0.16 | −0.59 to 0.26 |
| During- vs. pre-pandemic [43] | Patients | 7 | Cross-sectional-observational (p) (c) | 1,461 (p) 21,934 (c) | SMD | 0.48 | −0.08 to 1.04 |
| **Anxiety** | | | | | | | |
| During- vs. pre-pandemic [10] | Mixed | 52 | Longitudinal (within person) | | SMC | **0.13** | **0.02 to 0.23** |
| March–April 2020 vs. before pandemic | Mixed | 52 | Longitudinal (within person) | | SMC | 0.14 | −0.02 to 0.30 |
| May–July 2020 vs. before pandemic | Mixed | 52 | Longitudinal (within person) | | SMC | 0.05 | −0.04 to 0.14 |
| During pandemic vs. pre-pandemic [5] | General | 34 | Longitudinal (within person) or cross-sectional (if pre-pandemic available) | | % increase | **25.6** | **23.2 to 28.0** |
| During- vs. pre-pandemic[2] [59] | General | 10 | Longitudinal (within-person)/experimental (restrictions vs. no restrictions) | | Hedges' g | **0.18**[2] | **0.07 to 0.27** |
| During- vs. pre-pandemic [43] | General | 23 | Cross-sectional-observational (p) (c) | 49,746 (p) 132,145 (c) | SMD | **0.40** | **0.15 to 0.65** |
| During- vs. pre-pandemic [43] | HCW | 13 | Cross-sectional-observational (p) (c) | 5,508 (p) 22,204 (c) | SMD | −0.08 | −0.66 to 0.49 |
| During- vs. pre-pandemic [43] | Patients | 6 | Cross-sectional-observational (p) (c) | 1,845 (p) 12,458 (c) | SMD | 0.31 | −0.07 to 0.69 |
| During- vs. pre-pandemic [98] | Mixed | 4 | Longitudinal (>90% within-person) | 4,344 | SMD | **0.15**[1] | **0.07 to 0.22** |
| | Mixed | 1 | Longitudinal (>90% within-person) | 217 | PCD | −0.05[1] | −0.20 to 0.11 |
| Social restrictions vs. pre-pandemic/pre-PHSM [113] | General | 19 | Longitudinal, cross-sectional | | Cohen's d | 0.26 | −0.04 to 0.56 |
| **Mental health problems (non-specific)** | | | | | | | |
| During- vs. pre-pandemic [10] | Mixed (total) | 61 | Longitudinal (within person) | 55,015 | SMC | **0.11** | **0.04 to 0.17** |
| During- vs. pre-pandemic[2] [98] | Mixed | 12 | Longitudinal (within person) or cross-sectional (if pre-pandemic available) | 15,692 | SMD | **0.15**[1] | **0.12 to 0.18** |
| | Mixed | 12 | Longitudinal (within person) or cross-sectional (if pre-pandemic available) | 18,985 | PCD | −0.03[1] | −0.09 to 0.04 |
| During- vs. pre-pandemic [59] | General | 20 | Longitudinal (within-person), experimental (restrictions vs. no restrictions) | 72,004 | Hedges' g | **0.17** | **0.07 to 0.26** |
| Social restrictions vs. pre-pandemic/pre-PHSM [113] | General | 33 | Longitudinal, cross-sectional | | Cohen's d | **0.41** | **0.17 to 0.65** |

*(Continued)*

**Table 2.** (Continued)

| Variables | Population | Studies (*n*) | Designs of included studies | Pooled sample size | Metric | Pooled effect | 95% CI change/ increase |
|---|---|---|---|---|---|---|---|
| During- vs. pre-pandemic [10] | General | 75 | Longitudinal (within person) | | SMC | **0.12** | **0.04 to 0.19** |
| During- vs. pre-pandemic [10] | Preexisting physical | 14 | Longitudinal (within person) | | SMC | **0.25** | **0.07 to 0.43** |
| During- vs. pre-pandemic [10] | Preexisting mental | 25 | Longitudinal (within person) | | SMC | −0.02 | −0.21 to 0.18 |
| During- vs. pre-pandemic [10] | University students | 40 | Longitudinal (within person) | | SMC | 0.13 | −0.01 to 0.27 |
| During- vs. pre-pandemic [10] | Children/ adolescents | 38 | Longitudinal (within person) | | SMC | 0.11 | −0.03 to 0.26 |
| PHSM vs. pre-PHSM or pre-pandemic [89] | Children | 21 | Longitudinal and cross-sectional with retrospective pre-pandemic measures | 10,425 | Hedges' g | **0.28** | **0.15 to 0.41** |

[1] Deterioration for females compared to males.

[2] Anxiety and PTSD symptoms.

**Bold** represents significant effects.

(c), control participants; PCD, proportion change difference; PHSM, public health and social measures; (p), pandemic participants; SMC, standardized mean change; SMD, standardized mean difference; (n), equals number of studies or comparisons.

mobility decreased and daily SARS-CoV-2 infection rate increased [5] (Tables A and B in **S2 Text**).

Pooled prevalence rates based on cross-sectional data showed that above cut-off depression, anxiety, and PTSD symptom levels in the general population ranged respectively from 16% to 48%, from 15% to 47% and 9% to 33% (see **Fig 3** and **Table 3**). Despite some inconsistencies, the high heterogeneity of prevalence rates of depression, anxiety, and PTSD symptomatology across studies was partly explained by differences in assessment tools and cut-offs used [42,54,84,92] (Tables A, B, and C in **S2 Text**). Prevalence rates were often higher in females [50,77,102,106], in studies collected earlier in the pandemic [92,94], in younger age groups [9,92,102], in studies of lower quality or higher risk of bias [9,92,94,139,141], and in certain areas (e.g., European compared to Asian countries) [30,55,86,87,92]. In terms of COVID-19 exposure factors, prevalence rates of anxiety and PTSD symptoms were higher after peak of COVID-19 infections or when survey was taken closer to outbreak [33,138]. Anxiety prevalence was higher when public transportation was closed [30] and depression when government responded with more stringent measures [44].

**Healthcare workers.** No meta-analyses of longitudinal studies in healthcare workers were available. Based on during pandemic prevalence data and pre-pandemic comparative data, no significant differences in symptoms of depression and anxiety were found (SMD: −0.16, 95% CI: −0.59 to 0.26 and SMD: −0.08, 95% CI: −0.66 to 0.49, respectively, **Table 2**) and rates were not affected by COVID-19 patient contact [43]. As shown in **Table 3**, apart from some outliers, pooled cross-sectional prevalence of above cut-off depression, anxiety, and PTSD symptom levels during pandemic ranged from 19% to 42%, from 15% to 47%, and from 15% to 39%, respectively. Explanatory factors of the high heterogeneity and moderators were roughly similar to those in the general population such as different scales and cutoffs (e.g., [33]), non-random sampling or sample (size) differences (e.g., [34,36]), region of study (e.g., [74,135]), and quality or risk of bias scores of studies [9,42]. Similarly, prevalence rates were higher in studies with larger proportions of female versus male workers, medical versus non-medical professionals, frontline versus non-frontline workers, and nurses versus doctors (e.g., [27,41,56,97,102,105,107,123]). Prevalence rates of above cut-off PTSD level were related to

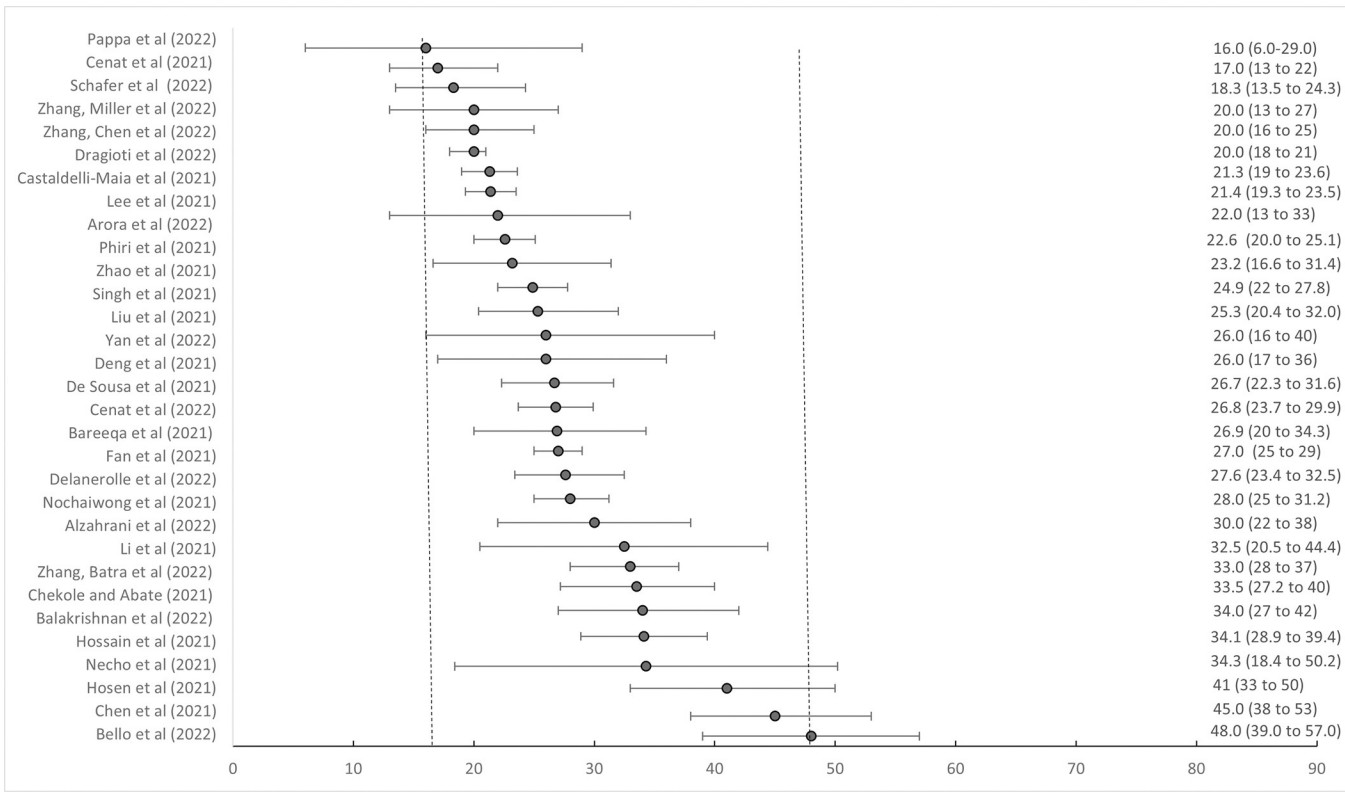

**Fig 3. Cross-sectional pooled prevalence rates of above cut-offs of symptoms of depression during COVID-19 pandemic in general populations.** Pooled prevalence rates from cross-sectional studies in general population with 95% confidence intervals.

COVID-19 mortality rate [60] and inconsistently to age (e.g., higher in older [61,143] or in younger health professionals [138]).

**People with preexisting physical or mental health disorders, or people infected with COVID-19.** A small but significant pre- to during pandemic increase in mental health symptoms in people with preexisting physical health conditions (SMC: 0.25, 95% CI: 0.07 to 0.43) was found, while for people with preexisting mental disorders, no such increase was found (SMC: −0.02, 95% CI: −0.21 to 0.18) [10] (**Table 2**). In a mixed population of patients with mental, physical, or COVID-19 diseases [43], depression and anxiety symptom levels were higher but not significantly different from those in matched studies with pre-pandemic prevalence data (SMD: 0.48, 95% CI: −0.08 to 1.04; SMD: 0.31, 95% CI: −0.07 to 0.69, respectively) [43] (**Table 2**). Apart from some outliers, pooled prevalence rates of depression, anxiety, and PTSD symptoms in COVID-19 patients ranged from 17% to 38%, from 23% to 40%, and from 15% to 42%, respectively (**Table 3**). In patients with physical diseases, pooled prevalence rates ranged from 31% to 37% and from 23% to 37% for depression and anxiety symptoms, respectively, over 50% in patients with eating disorders and roughly between 20% and 40% in perinatal women (**Table 3**). Although the high heterogeneity in patient populations remained often unexplained, some explanatory factors were different scales and cut-offs used (e.g., [48,50]) and sampling procedures (e.g., [35]). Higher prevalence rates of depression and anxiety in females (e.g., [102,111]), in certain regions of studies [102,111,120], in clinically severe, hospitalized and acute COVID-19 patients [35,124] and when COVID-19 mortality rates were higher [60]. High heterogeneity in during pandemic prevalence rates among perinatal women

**Table 3. Ranges of prevalence rates (of above cut-off scores) during pandemic from meta-analyses of pooled cross-sectional data.**

| Populations | Depression symptoms | | Anxiety symptoms | | PTSD symptoms | |
|---|---|---|---|---|---|---|
| | Pooled prevalence range | References | Pooled prevalence range | References | Pooled prevalence range | References |
| General populations | 16% to 48% | [9,29,30,33,38,42,44,50,54,55,67,77,84,85,88,91,92,94,100–102,106,123,132,136,139–141] | 15% to 47% | [9,29,30,33,38,42,45,50,54,55,77,79,84,85,87,91,92,94,100–102,106,111,123,133,136,139–141] | 9% to 33% | [9,38,50,55,60,61,64,76,77,85,86,91,100–102,138] |
| | *Outliers:* 14% and 65% | [37,91] | *Outlier:* 72% | [37] | 49% | [66] |
| Healthcare workers | 19% to 42% | [29,30,33,34,36,38,41,42,79,81,83,86,88,97,99–102,105–107,110,143] | 15% to 47% | [9,25,29,30,33,34,36,41,42,47,49,50,53,56,62,65–67,78,79,81,86,91,97,99–102,105–107,110,123,127,143] | 15% to 38% | [9,34,38,47,50,56,60,61,73,74,77,78,91,101,102,125,135,138,143] |
| **Patients:** | | | | | | |
| COVID-19 infection | 17% to 38% | [35,38,50,79,102,121,124,136] | 23% to 40% | [35,48,79,102,111,121,124] | 15% to 42% | [35,60,61,120,138] |
| | *Outlier:* 55% | [48] | *Outliers:* 14% and 64% | [50,136] | 94% | [102] |
| Somatic disorders | 31% to 37% | [26,102,114] | 23% to 39% | [102,111,114,115] | | |
| | *Outlier:* 17% | [115] | | | | |
| Eating disorders | 55% | [69,112] | 50 and 64% | [69,112] | | |
| Mixed disorders | 22% | [30] | | | | |
| Perinatal women | 23% to 34% | [31,39,68,70,82,96,102,131] | 17% to 40% | [31,39,68,70,102,111,144] | | |
| | *Outliers:* 17% and 40% | [68,144] | *Outlier:* 50% | [68] | | |
| Students | 23% to 39% | [27,28,30,32,40,46,51,71,79,88,103,104,109,118,123,129,130,142] | 28% to 44% | [28,32,46,71,79,90,103,104,109,111,116,119,128,130,142] | 30% | [27] |
| | *Outliers:* 50%,52%, 63%,65% | [79,90,106,119] | *Outliers:* 18% and 52%, 55% | [90,106,123] | | |
| Children/ adolescents | 22% to 29% | [52,63,93,95] | 21% to 34% | [28,52,63,93,122] | 28%, 48% | [52,137] |
| | *Outlier:* 42% | [122] | | | | |

Pooled prevalence rates with 95% confidence intervals from each review with meta-analyses of mental health outcomes in all populations can be found in the S2 Text (Tables A, B, and C in S2 Text).

was infrequently explained by region of study (lower in Asian than in western countries and higher in low- versus high-income countries [39,68,102]).

**Students.** A nonsignificant small increase from pre- to during pandemic in mental health problems (including anxiety and depression) was found for university students (SMC: 0.13, 95% CI: −0.01 to 0.27) [10]. Apart from outliers, cross-sectional prevalence rates of above cut-off depression and anxiety symptoms in students ranged from 23% to 39% and from 28% to 44%, respectively, and prevalence for above cut-off PTSD level was 30% (**Table 3**). Prevalence rates were higher in certain regions (e.g., non-China) [71,109,130], in females [28,104], and for specific assessment tools [32,90,109].

**Children and adolescents.** During- compared to pre-pandemic mental health symptoms in children and adolescents were not significantly increased in a meta-analysis of exclusively longitudinal data (SMC: 0.11, 95% CI: −0.03 to 0.26) [10], but in children between 5 and 13 years of age, symptoms were significantly higher when based on longitudinal and cross-sectional compared to retrospective pre-pandemic data ([89]; SMC 0.28; 95% CI: 0.15 to 0.41) (**Table 2**). In children and adolescents combined, pooled prevalence of above cut-off depression and anxiety levels ranged from 22% to 29% and from 21% to 34%, respectively, and for above cut-offs PTSD symptom level 28% resp. 30% (**Table 3** and Tables A, B, and C in **S2 Text**). High heterogeneity was found [52,63,93,95,122] and subgroup analyses showed higher prevalence rates of depression and anxiety symptoms in studies with a higher proportion of girls/females and in adolescents compared to children. PTSD symptom prevalence rate was higher in children and adolescents from Northern America and Europe compared to Southeast Asia [52,63,93,137]. In only few reviews with meta-analyses, a pooled prevalence rate in caregivers (e.g., mothers of young children [126]) and in working populations (e.g., teachers [58,108]) was calculated (Tables A, B, and C in **S2 Text**).

## Discussion

### Summary of key findings

In this umbrella review, we narratively summarized outcomes of 123 systematic reviews with meta-analyses to assess the association between COVID-19 and symptoms of common mental health disorders. The few reviews that pooled longitudinal data consistently showed a small increase in symptoms of depression and anxiety (and partly PTSD) during the early pandemic compared to pre-pandemic periods in the general population [5,10,59]. The increase of depression symptoms was generally larger and longer lasting than for anxiety [5,10]. Strict measures compared to less social restrictions resulted in higher depression symptoms as well [113]. Subgroup analyses of pooled during- to pre-pandemic or pre-implementation of public health and social measures data showed that mental health symptoms deteriorated more for people with preexisting physical but not for people with mental health conditions [10]. During pandemic, depression and anxiety symptoms were not different in healthcare workers nor in subgroups of any patients (including COVID-19), but these findings were based on cross-sectional data compared to pre-pandemic matched data [43]. Although the high heterogeneity between studies with pre- and during pandemic data could not be explained by differences in age and sex or gender in some reviews [10,59], other reviews did show that mental health of females and younger age groups including children between 5 and 13 years was more affected by the pandemic or by social restrictions [5,43,89,98,113]. Similar discrepancies were found in terms of country-level or COVID-19 exposure factors with some meta-analyses showing no evidence that factors such as continent, COVID-19 case/death rate, or economic situation explained the heterogeneity between longitudinal studies of general populations [10,59]. Others however found that region (e.g., Europe versus Southeast Asia) decreased human mobility

and higher daily SARS-CoV-2 infection rates were associated with depression and anxiety symptoms during the pandemic or during social restrictions [5,113].

Pooled cross-sectional data from general and specific populations and healthcare workers indicated wide ranges of prevalence rates. High heterogeneity was often explained by assessment tools and cut-offs used, sampling procedures, and quality of reviews. Also, higher rates were found in females, in certain regions, in acute or clinically severe COVID-19 patients, in frontline versus non-frontline healthcare workers and related to some COVID-19 exposure factors such as mortality rate. However, these findings need to be interpreted with caution and no causal inferences can be made due to the cross-sectional designs of the studies pooled in these meta-analyses.

### Interpretation of findings

The small increases of depression and anxiety symptoms are in line with population-based studies showing peaks of symptom prevalence during implementation of public health measures and social restrictions [145,146] and with recent meta-analytic findings of lower self-reported mental health in the first 2 months of the pandemic [147]. However, our umbrella review findings are in contrast with recent studies from local (mostly Northern European) countries that found no change or a decrease in during- compared to pre-pandemic mental health disorders and symptoms, e.g., for depression based on diagnostic interviews, or only an increase in subgroups of females or younger age groups [148,149]. Sampling and measurement differences may explain these discrepancies. For example, reviews pooling longitudinal symptom level data from multiple studies perhaps picked up more subtle nonclinical changes on self-report measures than individual longitudinal studies with outcomes based on clinical interviews. We also conclude, in line with the pooled longitudinal data in the reviews of Cénat and colleagues and Salanti and colleagues [92,147], that largest increases of mental health symptoms such as depression took place in the early phases of the pandemic [10] and during periods of social restrictions [113]. In May to July 2020, the SMD for depression was however still increased and only marginally lower than in March to April 2020 [10] and this is not in line with the continuous decline in anxiety and depression reported by Cenát [92] and Salanti [147]. An explanation may be that the more recent reviews that included studies with longer follow-up times were able to capture long-term trends of symptoms during the pandemic [92,147] specifically among more vulnerable individuals, such as females and young people [7].

Important issues to address in our umbrella review are the very high heterogeneity scores between studies (>90%) [10,59,113] and the lack of assessment and interpretation of risk of bias of primary studies [10,43,113], as this may compromise the certainty of the evidence. Across reviews, methodological and individual-level factors such as assessment tools used or age and sex or gender, explained some of the heterogeneity although inconsistently. Also, COVID-19 exposure factors such as daily COVID-19 cases and mobility indices and strictness of social restrictions, were associated with increases of depression and anxiety symptom levels in some reviews of longitudinal [5,113] and cross-sectional studies [30,44,60]. This is in line with the longitudinal study of Aknin and colleagues and the meta-analysis of Salanti and colleagues [147,150] but in contrast with other reviews of longitudinal data [10,59]. The Bayesian meta-regression and meta-analysis approaches in the Santomauro and colleagues and Salanti and colleagues reviews that use additional informative data (e.g., from cross-sectional samples or during pandemic longitudinal data) are more powerful and this may explain the differences with other reviews of longitudinal data with conventional random-effects meta-analysis. Still, heterogeneity in most reviews was high and largely unexplained. This may be due to the sparse

assessment of multiple COVID-19 exposure or individual-level factors such as economic support or situation on mental health symptoms in longitudinal studies. Also, COVID-19 exposure factors such as numbers of cases and stringency of the measures are strongly correlated over time in terms of their influence on mental health outcomes and difficult to disentangle [5,10]. This issue with the COVID-19 exposure factors calls for a more integrative approach to examine the inter-relatedness of several social, economic, and behavioral factors that may explain the association between public health measures and social restrictions during pandemics and mental health [113,151].

That females and younger age groups experienced a larger deterioration in mental health during the pandemic is a rather consistent finding across reviews presented here and in line with recent longitudinal studies with national representative or probabilistic samples [4,146]. In students, however, no significant worsening of mental health was found [10]. During pandemic, worsened or sustained deterioration of anxiety and depression has however been found in students with feelings of loneliness during the pandemic [152], indicating individual variation of impact of the pandemic even within specific subgroups.

Furthermore, in healthcare workers and in a mixed patient population, the during pandemic mental health symptoms were not significantly higher compared to matched pre-pandemic data [43]. It needs to be emphasized here that longitudinal within-person during- and pre-pandemic data were lacking among health workers and mixed patient populations, impeding strong conclusions regarding the pandemic-related changes in mental health in these groups. However, subgroup findings of cross-sectional prevalence rates cautiously suggested that pandemic exposure factors (e.g., COVID-19 mortality rate, region) negatively affected specific workers and patients (i.e., females, nurses, younger workers, or more severely infected patients). This is in line with recent individual longitudinal studies in these populations showing that anxiety among healthcare workers including females and young people was increased when exposure rates were highest, particularly shortly before a phase of implementing public health and social measures [7,153,154].

That the mental health of people with preexisting physical diseases was significantly deteriorated during the pandemic [10], is in line with other longitudinal studies and explained by higher loneliness and isolation scores [7,145]. However, that patients with mental disorders showed no such deterioration [10], seems inconsistent with findings indicating greater vulnerability in people with a history of mental disorders at least on the longer run during the pandemic [7]. This inconsistency may be explained by several phenomena. One is that pandemic mental health indicators like suicidality, although outside the scope of this review, tend to decrease in the initial phase of a disaster when people are less self-focused but may rise again when the situation is normalized and more long-term negative (socioeconomic) consequences become apparent [155]. This inconsistency may also be explained by great individual variation among people with mental disorders, with pandemic or implementation of public health and social measures related increases of anxiety and depression symptoms for some but beneficial effects for others (e.g., due to reduced social pressures). A ceiling effect may play a role as well, with symptoms already being that high that large increases cannot be expected [156]. That people with physical and mental disorders were less reactive to social restrictions in terms of depression symptoms than people who had no such disorders [113] may be explained by resilient adaptation and less loneliness in people with chronic somatic diseases during later phases of implementing public health and social measures [157] and by the less reactive peaks in mental health during public health and social measures in people with mental disorders [145].

For the association between the pandemic and mental health in other vulnerable groups from different specific sociodemographic backgrounds, meta-analyses of longitudinal data were lacking [10,43]. Findings are inconsistent, with some longitudinal studies suggesting that

people with ethnic diverse backgrounds, low level of education, and financial difficulties showed longer-lasting poorer mental health during the pandemic [7], while other studies indicate significant mental health deterioration in all sociodemographic groups [146].

### Implications of the findings

The findings presented here have implications for mental health researchers, policy makers, and public health professionals involved in current and future public health crises. First, more powerful meta-analysis methods with, e.g., Bayesian approaches, individual participant data from multiple countries, and linear and nonlinear assessments of change, should be employed to more accurately plot mental health trajectories. Second, individual-level socioeconomic and pandemic exposure factors including time as continuous factor (instead of estimations based on time-intervals or averages [5,59]) should be assessed more comprehensively and accurately. Third, accurate exposure trackers such as the OXFORD COVID-19 Government Response Stringency and Google mobility indices should be linked to longitudinal within-person data from probabilistic or national representative samples [11]. Finally, policy interventions could be developed in such a way that it includes repeated and systematic data collection in vulnerable groups, for example, through population panels. Since a digital and tele-mental health revolution unfolded during the pandemic [158], these developments may be incorporated in remote mental healthcare interventions making monitoring and prevention and treatment scalable and cost-effective socioeconomic approach for vulnerable individuals [159].

### Strengths and limitations

The broad scope of our search, rigorous rating of methodological quality, and complete overview of the evidence from available systematic reviews and meta-analyses until August 12, 2022, are clear strengths of our umbrella review. Certain limitations need to be taken into account as well. First, because we summarized the pooled prevalence rates and SMCs and the heterogeneity metrics and subgroup analyses from each individual meta-analysis qualitatively, quantitative reduction of inconsistencies or biases across reviews was lacking. Second, very few reviews pooled within-person data from multiple longitudinal studies and almost all pooled cross-sectional data from validated screening instruments often with different cut-offs for case-finding. These are unavoidable limitations inherited from the source reviews but warrant caution when interpreting findings as a cause-and-effect relationship to the pandemic. Importantly, prevalence rates are often higher in studies with more females and lower in studies from Asia irrespective of the pandemic [160] and screening instruments with different cut-offs for case-finding often reflect only a mild level of symptomatology of a disorder with short-term duration [161,162]. Third, we did not calculate the overlap between individual studies included in the systematic reviews with meta-analyses of our umbrella review. This potential for non-independence in the primary studies across similar reviews [163], may however be less problematic when analyzing outcomes from reviews in a narrative instead of a quantitative way, and the meta-analyses with pooled longitudinal data were inherently different in nature making overlap less likely. Finally, findings synthesized from meta-analyses presented here do not equally represent all regions across the world with a lack of studies from LMICs, making it difficult to infer conclusions in terms of mental health in relation to COVID-19. In resource-limited settings, however, people suffer from the greatest burden of mental illness although treatment and prevention of mental health problems is often lacking [164].

## Conclusions

Evidence suggests increased prevalence of mental health outcomes, particularly depression symptoms, during the pandemic and during implementation of public health measures and social restrictions, predominantly in young people, females, and people with chronic somatic disorders. Pooled prevalence rates of common mental health symptoms during the pandemic ranged from about 10% to 50% but lacked pre-pandemic comparison and true clinical value. At the time of writing this article, the COVID-19 pandemic in most regions worldwide has subsided and data collection of COVID-19 cases and deaths has been discontinued. In some areas, however, COVID-19 infection rates are still elevated, may increase again or new variants may keep emerging. The implications mentioned in our review, in terms of research and clinical and policy interventions, such as the implementation of scalable and widely accessible (remote) psychological interventions, are thus still timely and may offer more sound and definite policy directives to mitigate the impact of global public health disasters on mental health.

## Supporting information

**S1 PRISMA checklist. Prisma 2020 checklist.**
(DOCX)

**S1 Text. Search strategy, data extraction protocols, quality assessment. Table A**. Search strategies used to retrieve papers from different databases. **Table B.** List of excluded meta-analyses by full-text screening with exclusion reason.**Table C.** AMSTAR 2 ratings.
(DOCX)

**S2 Text. Results from reviews on prevalence of depression, anxiety, and PTSD symptoms during COVID-19. Table A.** Pooled prevalence rates and changes in depression symptoms, with heterogeneity scores and main subgroup findings. **Table B.** Pooled prevalence rates and changes in anxiety symptoms, with heterogeneity scores and main subgroup findings. **Table C**. Pooled prevalence rates and changes in PTSD symptoms, with heterogeneity scores and main subgroup findings. **Fig 1.** PRISMA flow chart initial search December 31, 2019 until October 6, 2021. **Fig 2.** PRISMA flow chart updated search October 7, 2021 until August 12, 2022.
(DOCX)

## Acknowledgments

We like to thank Ms. Zhuoli Zhang (Catholic University of Leuven, Belgium) for her help in the screening and selection process and data-extraction from full-text articles retrieved from the updated search.

The authors alone are responsible for the views expressed in this publication; they do not necessarily represent the decisions, policy, or views of the WHO, the European Community, or institutions with which the authors are affiliated.

## Author Contributions

**Conceptualization:** Anke B. Witteveen, Pim Cuijpers, José Luis Ayuso-Mateos, Corrado Barbui, Brandon Gray, Mark van Ommeren, Marianna Purgato, Marit Sijbrandij.

**Data curation:** Susanne Y. Young, Maria Cabello, Camilla Cadorin, Naomi Downes, Daniele Franzoi, Michael Gasior, Christina Palantza, Judith van der Waerden, Siyuan Wang.

**Formal analysis:** Anke B. Witteveen, Federico Bertolini, Maria Cabello, Camilla Cadorin, Naomi Downes, Daniele Franzoi, Christina Palantza, Judith van der Waerden, Siyuan Wang.

**Funding acquisition:** Anke B. Witteveen, Pim Cuijpers, José Luis Ayuso-Mateos, Corrado Barbui, Marianna Purgato, Marit Sijbrandij.

**Investigation:** Anke B. Witteveen, Susanne Y. Young, Federico Bertolini, Maria Cabello, Camilla Cadorin, Naomi Downes, Daniele Franzoi, Judith van der Waerden, Siyuan Wang.

**Methodology:** Anke B. Witteveen, Pim Cuijpers, Federico Bertolini, Marit Sijbrandij.

**Project administration:** Susanne Y. Young, Christina Palantza.

**Resources:** Susanne Y. Young, Maria Cabello, Camilla Cadorin, Michael Gasior, Christina Palantza.

**Supervision:** Anke B. Witteveen, Susanne Y. Young, Pim Cuijpers, Brandon Gray, Maria Melchior, Mark van Ommeren, Marianna Purgato, Marit Sijbrandij.

**Writing – original draft:** Anke B. Witteveen.

**Writing – review & editing:** Anke B. Witteveen, Pim Cuijpers, José Luis Ayuso-Mateos, Corrado Barbui, Federico Bertolini, Maria Cabello, Camilla Cadorin, Naomi Downes, Daniele Franzoi, Michael Gasior, Brandon Gray, Maria Melchior, Mark van Ommeren, Christina Palantza, Marianna Purgato, Judith van der Waerden, Siyuan Wang, Marit Sijbrandij.

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
