## [Editor Report · Decision Letter 0]

26 Jul 2022

Dear Dr Witteveen, 

Thank you for submitting your manuscript entitled "Impact of COVID-19 on common mental health symptoms in the early phase of the pandemic: an umbrella review of the evidence" for consideration by PLOS Medicine.

Your manuscript has now been evaluated by the PLOS Medicine editorial staff and I am writing to let you know that we would like to send your submission out for external peer review.

Your manuscript is currently under consideration as part of the Special Issue on the COVID-19 pandemic and global mental health. The deadline for the Special Issue is being extended to December 15 2022, with anticipated publication in Q1 2023 (subject to change dependent on submission volume). We intend to publish all papers accepted for the Special Issue simultaneously.

Given that this extension was announced after you submitted your manuscript for consideration, we appreciate that you may no longer wish for your manuscript to considered specifically for the Special Issue. If this is the case, or if you have any other questions, please feel free to contact me (cdavidson@plos.org) and this can be discussed.

Before we can send your manuscript to reviewers, we need you to complete your submission by providing the metadata that is required for full assessment. To this end, please login to Editorial Manager where you will find the paper in the 'Submissions Needing Revisions' folder on your homepage. Please click 'Revise Submission' from the Action Links and complete all additional questions in the submission questionnaire.

Please re-submit your manuscript within two working days, i.e. by Jul 28 2022 11:59PM.

Kind regards,

Callam Davidson

Associate Editor

PLOS Medicine

---

## [Decision Letter · Decision Letter 1]

31 Oct 2022

Dear Dr. Witteveen,

Thank you very much for submitting your manuscript "Impact of COVID-19 on common mental health symptoms in the early phase of the pandemic: an umbrella review of the evidence" (PMEDICINE-D-22-02427R1) for consideration at PLOS Medicine. 

Your paper was evaluated by an associate editor and discussed among all the editors here. It was also discussed with an academic editor with relevant expertise, and sent to independent reviewers, including a statistical reviewer. The reviews are appended at the bottom of this email and any accompanying reviewer attachments can be seen via the link below:

[LINK]

In light of these reviews, I am afraid that we will not be able to accept the manuscript for publication in the journal in its current form, but we would like to consider a revised version that addresses the reviewers' and editors' comments. Obviously we cannot make any decision about publication until we have seen the revised manuscript and your response, and we plan to seek re-review by one or more of the reviewers. 

We expect to receive your revised manuscript by Nov 21 2022 11:59PM. Please email us (plosmedicine@plos.org) if you have any questions or concerns.

We look forward to receiving your revised manuscript. 

Sincerely,

Callam Davidson, 

PLOS Medicine

plosmedicine.org

Please update your title to ‘COVID-19 and common mental health symptoms in the early phase of the pandemic: an umbrella review of the evidence’.

Please report your abstract according to PRISMA for abstracts, following the PLOS Medicine abstract structure (Background, Methods and Findings, Conclusions) http://www.plosmedicine.org/article/info:doi/10.1371/journal.pmed.1001419

Abstract Methods and Findings:

* Please ensure that all numbers presented in the abstract are present and identical to numbers presented in the main manuscript text.

* Please quantify the main results (with 95% CIs and p values).

Please include continuous line numbering throughout your manuscript. 

We require that systematic reviews are updated to within roughly 6 months of the expected publication date. Please update your search to the present time.

Please temper claims to primacy (‘no comprehensive overview…has been conducted’) by including ‘to our knowledge’ or similar. 

Thank you for including a PRISMA checklist. Please cite the checklist in your methods as ‘S1 checklist’ (or similar), and also update the checklist to use section headings/paragraph numbers rather than page numbers. 

Please provide titles and legends for all figures and Tables (including those in Supporting Information files).

Comments from the reviewers:

Reviewer #1: This is an interesting umbrella review on the impact of COVID-19 on common mental health symptoms in the early phase of the pandemic. However, there are a few issues needing attention.

1) The main result of an increased prevalence of mental health symptoms during the pandemic compared to before is well known and widely studied and reported. Many other studies even went on to later phase of the pandemic until 2022 and provided further insight of mental health problems in the populations throughout the pandemic. Now the question is what's novelty of this study? Any new insight not being discovered before?

2) This umbrella review approach claims to address the issues like inconsistencies, biases and gaps of knowledge. Have the authors achieved all these in the paper and given better and more consistent results? Particularly the very high heterogeneity score is throughout all the pooled analyses, which means there are still inconsistencies and heterogeneities among the pooled studies. In short, have we seen the inconsisitency and bias reduced by this umbrella review quantatively and by what measure?

3) Public health and social measures (PHSMs). What exacly are PHSMs? Have the authors considered the impact of different lock down measures and durations on mental health symptoms such as strict lockdown in China and relatively soft lockdown in UK? How about financial support in each country to relieve the mental health symptoms? These are just a few factors. Without detailed subgroup analyses on these important factors, the pooled results are unlikely to be consistent and less biased.

4) Table 2 is a bit confusing. Should it be during pandemic vs before rather than before vs during pandemic? So far, it looks like there are more depression and anxiety before pandemic.

Reviewer #2: Many thanks for your kind invitation to review the manuscript titled: ''Impact of COVID-19 on common mental health symptoms in the early phase of the pandemic: an umbrella review of the evidence''

This is a well designed and well executed umbrella review of the literature and I would like to commend the authors for their work. 

Despite its numerous limitations, I believe it will make a significant contribution to the literature and shed some light in the relatively 'muddy' and often contradictory findings around the impact of COVID-19 on mental health worldwide.

I only suggest some minor points for revision:

* The finding that healthcare workers' mental health was not greatly affected when comparing pre- and during the pandemic needs to be further interpreted. It would be important if the authors suggested some explanations for this (?unexpected) phenomenon

* The same stands for people with pre-existing mental health problems. It would be important if the authors expanded on the potential reasons of why this group was not particularly affected by Covid-19 (e.g. the 'pulling together' phenomenon has been suggested for the non-increase in serious mental illness during the initial phases of the pandemic)

* I was wondering whether there was any data collected or any comment that could be made on suicidality of the groups investigated ?

* Finally, although the authors indicate that one of the study's aims is 'to identify targets for clinical- and policy interventions' , this is not done adequately in my opinion. It is suggested that the authors add a separate, distinct, small paragraph on policy interventions and future implications of their study's findings 

Reviewer #3: Thank you for the opportunity to review this umbrella review. This study represents an important contribution to the literature. I have only a few comments:

1) The authors are entirely correct in my view tor prioritise reviews with meta-analyses focused on longitudinal data or studies with comparative pre-pandemic prevalence data. However, the rationale for prioritising this type of study needs to be better motivated in the introduction, expanding on the point that "cross-sectional primary studies [...] do not allow any causal inferences to the pandemic". For instance, highlighting that high prevalence estimates during the pandemic are not meaningful in the absence of comparative data collected with methodological consistency (most crucially sampling), the importance of probability-based sampling when generating prevalence estimates (e.g. Pierce, M., McManus, S., Jessop, C., John, A., Hotopf, M., Ford, T., ... & Abel, K. M. (2020). Says who? The significance of sampling in mental health surveys during COVID-19. The Lancet Psychiatry, 7(7), 567-568.), and the advantages of examining the same people over time when representative data is unavailable or the comparability of sampling is compromised. Essentially including a paragraph motivating why the review views studies with comparative pre-pandemic data as at lower risk of bias. 

2) Please amend the below point in the introduction. Sun et al. include convenience samples in their 'general population' samples (e.g. Benz et al, Katz et al.) and I don't see subgroup analyses of 'population-based' studies. There are also only a handful of general population studies examined in Sun et al. When a larger number of longitudinal studies with representative sampling are examined an increase in mental health symptoms in the initial stages of the pandemic is found (see Robinson et al. review Table 2 https://doi.org/10.1016/j.jad.2021.09.098). 

"Although findings from population-based studies in the initial stages of the pandemic indicate that most people were resilient and did not experience increases in distress [4]"

Sun Y, Wu Y, Bonardi O, Krishnan A, He C, Boruff JT, et al. Comparison of Mental Health

Symptoms prior to and during COVID-19: Evidence from a Living Systematic Review and Metaanalysis. [cited 23 Jun 2022]. doi:10.1101/2021.05.10.21256920

3) When discussing results from Table 2 it may make more sense to highlight the design of the studies included (exclusively longitudinal studies vs. estimates including comparisons to different pre-pandemic samples) as this may be a key factor in explaining discrepancies between estimates (e.g. point estimates of .67 vs. .23 differ substantially below with no indication how the studies they are based on differ in their design). Currently, how these findings are discussed seems contradictory in places (e.g. the prevalence of anxiety was higher in the pandemic by 0.40 SMD but not significant over time, SMD = 0.05 in May-July) but is most likely a function of the types of studies estimates are based on. 

"Table 2 presenting selected outcomes from three reviews with meta-analyses of longitudinal data or comparative pre-pandemic prevalence data, increased rates of depression symptoms were found (SMD: 0.67, 95% CI: 0.07 to 1.27 [34]). Differences in depression symptoms remained significant over time in the first half year of 2020 (March-April SMC: 0.23, 95% CI: 0.11 to 0.34 and May-July SMC: 0.20, 95% CI: 0.10 to 0.30)[11]." 

4) Discussion - increases in mental health symptoms during the public health and social measures need to be contextualised in relation to studies explicitly estimating the link between COVID-19 policy stringency and mental health measures which have found this link to be small - Aknin, L. B., Andretti, B., Goldszmidt, R., Helliwell, J. F., Petherick, A., De Neve, J. E., ... & Zaki, J. (2022). Policy stringency and mental health during the COVID-19 pandemic: a longitudinal analysis of data from 15 countries. The Lancet Public Health, 7(5), e417-e426.

5) Discussion -please note where the below result can be found in the results section:

"Meta-regression analyses showed higher prevalence rates of mental health symptoms predominantly in early pandemic as well as an increasing trend between January to September 2020"

6) The authors may be over-interpreting trends in cross-sectional estimates which is a concern particularly given the point on this data not allowing causal inferences made in the introduction. For instance, many studies have found age-differences in distress and common mental health symptoms to be typical immediately prior to the pandemi (e.g. for the UK Daly, M., Sutin, A. R., & Robinson, E. (2020). Longitudinal changes in mental health and the COVID-19 pandemic: Evidence from the UK Household Longitudinal Study. Psychological medicine, 1-10.). The fact that there are also age differences in mental health symptoms in (cross-sectional) data collected during the pandemic may say nothing about the effect of the pandemic. Also, an increasing trend in cross-sectional prevalence estimates could be driven by differences in sample composition or representativeness over time. Can the authors address this point?

7) Please include page/line numbers. 

8) Concluding section summarises the trends identified v.well. 

Reviewer #4: This article presents an umbrella review of systematic reviews and meta-analyses of the impact of the early phase of the COVID-19 pandemic on mental health. The authors identified 58 suitable reviews that were of variable quality according to standard criteria, with only one being rated as very high quality. Estimates of depression and anxiety during 2020 were summarized in different populations including the general population, healthcare workers, people with pre-existing mental and physical ill-health, young people, etc. The authors conclude that anxiety and depression were increased during these months in the general population and among those with physical health problems, but not in other groups. Mental health was also more impaired during periods during which restrictive public health and social measures (PHSMs) were implemented compared with periods without PHSMs. 

1. This review is a major undertaking, and the approach is very thorough. It is striking just how many individual reviews were published within a few months of the onset of the pandemic. 577 reviews passed initial selection procedures, although this was whittled down to 58. Given that COVID-19 was unknown in most countries until 2020, this is an astonishing number. One has the impression that almost as many reviews as primary articles were published over this period. The authors rightly point out the importance of studies that included pre-pandemic data so that change in mental health could be examined.

2. The main findings are summaries of the meta-analyses and selective presentations of individual findings. They show considerable heterogeneity. It could be made clearer how some conclusions are drawn. For example, the conclusion that healthcare workers showed no rise in anxiety and depression during the early months of the pandemic appears to be based on a single meta-analysis (Kunzler et al, 2021), but tables D and E list numerous other reviews that drew different conclusions. This is only one comparison, but needs clarification.

3. Some clarification is also needed of the term 'young people'. Do the authors mean young adults, or does this group include children? Was there a lower age limit on the studies considered? 

4. One of the authors' conclusions is that people with pre-existing physical health conditions were at increased risk for mental ill-health. It is not clear from the description here whether this category included people with COVID-19 infection, or only those with a long-term condition unrelated to the virus. I am also confused by the conclusion that people with pre-existing physical health conditions had a higher prevalence, but that 'patients (including those with COVID-19 infections)' did not. Surely these 'patients' had pre-existing health conditions?

5. The authors emphasise that mental health was poorer during periods of PHSMs, and give the impression that PHSMs contributed to this rise. There are two issues here that are not adequately taken into account. First, the primary source for this finding (the meta-analysis by Prati & Mancini, 2021) did not compare periods during the COVID-19 pandemic when PHSMs were in force and not in force, but compared mental health pre-pandemic vs pandemic. Thus their conclusion is really that mental health was poorer during the pandemic compared with before. The way to evaluate the impact of PHSMs is to compare months during the pandemic that had higher or lower restrictions in the relevant country (using something like the Oxford Stringency index). This has been done in more recent studies beyond the scope of the reviews included here, though I'm not aware of meta-analyses in this area. Second, PHSMs typically coincide with periods of greater COVID-19 infection in the population, and this itself likely leads to increases in anxiety and depression. I don't think it is possible to infer that PHSMs themselves are responsible for any changes in mental health from the findings of present study; the Abstract and Conclusions should be modified accordingly.

6. Finally, the authors recognise that the reviews included in their coverage are limited to the early months of the pandemic globally. No mention is made of changes in the incidence of COVID infection, the likely impact of vaccination, and the course of the pandemic in 2021 and 2022. How important are the changes in mental health detailed here in the long run? The authors plead for more research particularly in low and middle income countries, but how relevant will new studies be at this stage, given the trajectory of the pandemic? A stronger argument needs to be made as to why these findings are not only of historical interest, but matter now.

[LINK]

---

## [Decision Letter · Decision Letter 2]

10 Feb 2023

Dear Dr. Witteveen,

Thank you very much for re-submitting your manuscript "COVID-19 and common mental health symptoms in the early phase of the pandemic: an umbrella review of the evidence" (PMEDICINE-D-22-02427R2) for review by PLOS Medicine.

I have discussed the paper with my colleagues and the academic editor and it was also seen again by three reviewers. I am pleased to say that provided the remaining editorial and production issues are dealt with we are planning to accept the paper for publication in the journal.

[LINK]

We look forward to receiving the revised manuscript by Feb 17 2023 11:59PM.   

Sincerely,

Callam Davidson, 

Associate Editor 

PLOS Medicine

plosmedicine.org

Requests from Editors:

Given the observational nature of much of the data included in the review, I would suggest avoiding terms that imply causality (e.g., ‘impact’, ‘effects’) and opt instead for ‘associations’, or similar. 

Thank you for updating your search period. Your search was updated to August 2022 – given that our decision letter was sent on October 31, 2022, we had anticipated that your search would be updated to October 2022. As previously noted, we ask that systematic review/meta-analyses updated to within 6 months of the date of publication. Please provide the rationale behind updating the search to August 2022 rather than October 2022.

Please confirm whether ‘gender’ is the appropriate term to be used in your study (as opposed to ‘sex’). The following article from the WHO may be useful - https://www.who.int/europe/health-topics/gender

Thanks for providing an Author Summary. I have attached suggested changes. Please implement these as appropriate and then insert the Author Summary into the main text (between the Abstract and the Introduction).

Please ensure changes in the analysis (including any deviations from the a priori protocol and those made in response to peer review comments) are identified as such in the Methods section of the paper, with rationale.

I could not access Figure 1, please confirm the file type is correct and the file is not corrupted. 

Please confirm throughout that in-text Tables/Figures are cited correctly.

Please carefully check your References for unnecessary details (e.g., publisher). References should be Vancouver style, further guidance can be found at: https://journals.plos.org/plosmedicine/s/submission-guidelines#loc-references

Similarly, some references are missing key information (e.g., 148, Salanti et al., is missing the journal and contains date of citation).

The last paragraph of the Conclusions section appears to repeat some information from the previous paragraph.

Please remove the Author contributions section from the main text (this information will be captured as metadata during the submission process.

Comments from Reviewers:

Reviewer #1: Thanks authors for their great effort to improve the manuscript. The authors have comprehensively addressed all my concerns. I am satisfied with the response and revision. No further issues needing attention.

Reviewer #3: All my comments have been addressed. The article has been strengthened substantially with the update of the review and introduction of additional discussion content. There were some small typos (e.g. line 65 no 'the' at start, line 70 'hence' does not work here, lines 457-459 are repeated in 466-468) the authors will need to proof read the paper well prior to publication. 

Reviewer #4: The authors should be congratulated on the hard work they have put into this revision. I am satisfied with the amendments made, and do not have any further specific recommendations for revision. However, I must say that the Abstract is a pretty dense and quite difficult read, and this may reduce the impact of the paper.

[LINK]

---

## [Editor Report · Decision Letter 3]

21 Feb 2023

Dear Dr Witteveen, 

On behalf of my colleagues and the Academic Editor, Dr Lola Kola, I am pleased to inform you that we have agreed to publish your manuscript "COVID-19 and common mental health symptoms in the early phase of the pandemic: an umbrella review of the evidence" (PMEDICINE-D-22-02427R3) in our upcoming Special Issue.

Please also note that there are remaining issues with the format of the references in your manuscript which will need to be corrected prior to publication. Notably, the inclusion of [Internet] and an associated URL is only required for items that can only be accessed via this route (i.e., this is not required for indexed journal articles with an associated DOI). Please see our website for instructions as to how references ought to be formatted: https://journals.plos.org/plosmedicine/s/submission-guidelines#loc-references

PRESS

We ask that you take this opportunity to read our Embargo Policy regarding the discussion, promotion and media coverage of work that is yet to be published by PLOS. As your manuscript is not yet published, it is bound by the conditions of our Embargo Policy. Please be aware that this policy is in place both to ensure that any press coverage of your article is fully substantiated and to provide a direct link between such coverage and the published work. For full details of our Embargo Policy, please visit http://www.plos.org/about/media-inquiries/embargo-policy/.

SPECIAL ISSUE PUBLICATION

We are intending to publish the Special Issue in April 2023. We appreciate that some authors may wish to begin publicising their work before the Special Issue launches. For this reason, we have decided to offer optional Early Article Posting for authors who feel that this would be beneficial.

Posting an early version of the article online removes the press embargo and allows authors to begin coordinating their own/institutional press activities, though it is important to note that this can result in reduced media coverage overall (we have published a related article on this in the context of preprints: https://theplosblog.plos.org/2020/05/preprints-and-the-media-a-change-to-how-plos-handles-press-for-papers-previously-posted-as-preprints/).

If you feel that your article may benefit from early versioning, please reach out to me (cdavidson@plos.org) and I will be happy to discuss this further. If we do not hear otherwise, we will assume you are happy to remain opted out of the early versioning process.

REPRODUCIBILITY

Sincerely, 

Callam Davidson 

Associate Editor 

PLOS Medicine